# Improving Neural Ordinary Differential Equations with Nesterov's Accelerated Gradient Method

**Nghia H. Nguyen**[*]
FPT Software AI Center
Ha Noi, Vietnam
nghianhh1@fsoft.com.vn

**Tan M. Nguyen**[*]
Department of Mathematics
University of California, Los Angeles
tanmnguyen89@ucla.edu

**Huyen K. Vo**
FPT Software AI Center
Ha Noi, Vietnam
huyenvtk1@fsoft.com.vn

**Stanley J. Osher**
Department of Mathematics
University of California, Los Angeles
sjo@math.ucla.edu

**Thieu N. Vo**
Faculty of Mathematics and Statistics
Ton Duc Thang University, Ho Chi Minh City, Vietnam
vongocthieu@tdtu.edu.vn

## Abstract

We propose the Nesterov neural ordinary differential equations (NesterovNODEs), whose layers solve the second-order ordinary differential equations (ODEs) limit of Nesterov's accelerated gradient (NAG) method, and a generalization called GNesterovNODEs. Taking the advantage of the convergence rate $\mathcal{O}(1/k^2)$ of the NAG scheme, GNesterovNODEs speed up training and inference by reducing the number of function evaluations (NFEs) needed to solve the ODEs. We also prove that the adjoint state of a GNesterovNODEs also satisfies a GNesterovNODEs, thus accelerating both forward and backward ODE solvers and allowing the model to be scaled up for large-scale tasks. We empirically corroborate the advantage of GNesterovNODEs on a wide range of practical applications, including point cloud separation, image classification, and sequence modeling. Compared to NODEs, GNesterovNODEs require a significantly smaller number of NFEs while achieving better accuracy across our experiments.

## 1 Introduction

Dynamical systems have been recently integrated into deep neural networks for modeling high-dimensional data. The advantage of this approach is that well-developed mathematical modeling techniques from dynamical systems can be employed to improve neural networks. Along this research direction, the correspondence between residual networks, a popular class of neural networks with skip connections, and the numerical solution of ordinary differential equations (ODEs) have been vastly studied in [15, 57, 47, 4]. The resulting Neural ODEs (NODEs) model when taking the the discretization step to zero have shown great promises in a wide range of applications including scientific discovery [24, 62], irregular time series modeling [46, 5], mean-field games [48], and generative modeling [13, 61]. NODEs model the dynamics of hidden state $\boldsymbol{h}(t) \in \mathbb{R}^N$ in a neural

---

[*] Co-first authors. Please correspond to: nghianhh1@fsoft.com.vn or tanmnguyen89@ucla.edu or vongoc-thieu@tdtu.edu.vn

36th Conference on Neural Information Processing Systems (NeurIPS 2022).

network by an ODE

$$\frac{d\boldsymbol{h}(t)}{dt} = f(\boldsymbol{h}(t), t, \theta), \ \ \boldsymbol{h}(0) = h(t_0), \tag{1}$$

where function $f$ captures the dynamics and is chosen to be a neural network with parameters $\theta$ that are learned from the data. Starting from the input $h(t_0)$ at the initial time $t_0$, NODEs compute the output $h(T)$ at time $T$ by solving the Initial Value Problem in Eq. (1) for some time $T \geq t_0$. NODEs are trained by optimizing the loss $L(\boldsymbol{h}(T))$ between the prediction $\boldsymbol{h}(T)$ and the ground truth where the parameters $\theta$ are updated using the following gradient [42]

$$\frac{d\mathcal{L}(\boldsymbol{h}(T))}{d\theta} = \int_{t_0}^{T} \boldsymbol{a}(t) \frac{\partial f(\boldsymbol{h}(t), t, \theta)}{\partial \theta} dt, \tag{2}$$

where $\boldsymbol{a}(t) := \partial\mathcal{L}/\partial\boldsymbol{h}(t)$ is the continuous adjoint state, which satisfies the continuous adjoint equation

$$\frac{d\boldsymbol{a}(t)}{dt} = -\boldsymbol{a}(t) \frac{\partial f(\boldsymbol{h}(t), t, \theta)}{\partial \boldsymbol{h}}. \tag{3}$$

NODEs solve both the ODE Eq. (1) in its forward pass and the ODEs (2) and (3) in its backward pass using black-box numerical ODE solvers. The literature refers to this approach as either the continuous adjoint method or optimise-then-discretise [21]. The number of function evaluations (NFEs) that these solvers need in a single forward and backward pass is among the main factors that decide the computational efficiency of the model, i.e. how fast the model is. Unfortunately, in many applications, NODEs require high NFEs in both training and inference, especially when the error tolerances of the solvers are set to small values for obtaining high accuracy. Furthermore, the NFEs increase rapidly when training progresses. High NFEs deteriorate the efficiency of NODEs, reduce the accuracy of the trained model, and results in instability during training, making it difficult to scale up the models to large-scale tasks [14, 8, 30, 37, 11].

## 1.1 Contribution

We propose the Nesterov Neural ODEs (NesterovNODEs) that leverage the continuous limit of the Nesterov's accelerated gradient (NAG) descent [52] and the convergence rate $\mathcal{O}(1/k^2)$ of the NAG scheme to enhance NODE training and inference. Our contributions are four-fold:

1. We formulate the NesterovNODE that solves Nesterov ODEs, i.e. second-order ODEs with a time-dependent damping term, instead of first-order ODEs (1). To improve computational efficiency of the model, we convert these second-order ODEs into equivalent systems of first-order differential-algebraic equations that are solved in both forward and backward propagations of the NesterovNODE.

2. To eliminate the potential blow-up problem in training NesterovNODEs, we further develop the Generalized NesterovNODEs (GNesterovNODEs) by introducing skip connections [16] and gating mechanisms [18] into NesterovNODEs. In general, GNesterovNODEs form a wide class of neural differential equations which are represented by differential-algebraic systems and contain NesterovNODEs as a subclass.

3. We prove that the continuous adjoint equation used to compute the gradients for updating the parameters $\theta$ in a GNesterovNODE also follows a generalized Nesterov ODE. Thus, the NFEs in both forward and backward passes of GNesterovNODEs are significantly reduced, especially when the solvers are used with small error tolerances.

4. We prove that the spectrum of the GNesterovNODE is well-structured. This property of GNesterovNODEs helps alleviate the vanishing gradient issue during training and allows the model to capture long-term dependencies in the data.

We empirically demonstrate the advantages of the NesterovNODEs/GNesterovNODEs over the baseline NODE and the state-of-the-art neural ODE models including the Heavy Ball NODEs (HBNODEs), which solve the continuous limit of the heavy ball momentum accelerated gradient descent [60] on a wide range of applications including point cloud separation, image classification, and kinetic simulation. In all experiments, our proposed models achieve better accuracy and smaller NFEs than the baselines.

## 1.2 Organization

We structure this paper as follows: In Section 2, we review HBNODEs and NesterovODEs. In Section 3, we present the algorithm and analysis of the NesterovNODEs and GNesterovNODEs. We study the spectrum structure of the adjoint equations of NesterovNODEs/GNesterovNODEs to show that NesterovNODEs/GNesterovNODEs can learn long-term dependencies effectively in Section 4. In Section 5 and 6, we empirically validate the advantages of NesterovNODEs/GNesterovNODEs and analyze our models with ablation studies. We discuss related works in Section 7. The paper ends with concluding remarks. Proofs and additional experimental details are provided in the Appendix.

## 2 An Integration of Nesterov ODEs into NODEs

We first establish a connection between NODEs and gradient descent (GD), then review the Heavy Ball Neural ODEs (HBNODEs), and motivate the integration of Nesterov ODEs into NODEs.

**ODE limit of gradient descent and connections to NODEs** Gradient descent (GD) has been among the methods of choice in optimization and machine learning for training complex systems. Starting from initial point $\boldsymbol{x}_0 \in \mathbb{R}^d$, GD iterates as $\boldsymbol{x}_k = \boldsymbol{x}_{k-1} - s\nabla F(\boldsymbol{x}_k)$ with $s > 0$ being the step size in order to find a minimum of the function $F(\boldsymbol{x})$. Let $s \to 0$, we obtain the following ODE limit of the GD

$$\frac{d\boldsymbol{x}}{dt} = -\nabla F(\boldsymbol{x}_t). \tag{4}$$

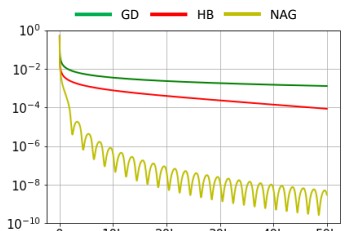

Comparing Eq. (1) and (4), we observe that a NODE solves the ODE limit of the GD where the gradient $-\nabla F(\boldsymbol{x}_t)$ is parameterized by a neural network $f(\boldsymbol{x}(t), t, \theta)$.

**Heavy ball neural ordinary differential equations** HBNODEs are proposed in [60]. This model takes advantage of the acceleration of heavy ball (HB) momentum [41] to reduce the NFEs needed for solving the ODEs and speed up NODEs. In particular, HBNODEs replace the first-order ODE limit of GD by the following second-order ODE limit of heavy ball momentum method:

$$\frac{d^2\boldsymbol{x}(t)}{dt^2} + \gamma \frac{d\boldsymbol{x}(t)}{dt} = -\nabla F(x(t)). \tag{5}$$

Figure 1: Comparing the convergence of GD, HB and NAG for solving the optimization problem $\min_{\boldsymbol{x}} F(\boldsymbol{x}) = \frac{1}{2}\boldsymbol{x}^\top \mathbf{L}\boldsymbol{x} - \boldsymbol{x}^\top \boldsymbol{b}$ where $\mathbf{L} \in \mathbb{R}^{d \times d}$ is the Laplacian of a cycle graph and $\boldsymbol{b}$ is a d-dimensional vector whose first entry is 1 and all the other entries are 0.

Similar to NODEs, [60] parameterize $-\nabla F(x(t))$ by a neural network $f(\boldsymbol{x}(t), t, \theta)$ and formulate HBNODEs as follows

$$\frac{d^2\boldsymbol{x}(t)}{dt^2} + \gamma \frac{d\boldsymbol{x}(t)}{dt} = f(\boldsymbol{x}(t), t, \theta), \tag{6}$$

where $\gamma > 0$ is the damping parameter, which can be a hyperparameter or a learnable parameter.

**Nesterov's accelerated gradient (NAG) momentum** Even though HB improves the convergence and accelerates GD, both GD and HB share the same convergence rate of $O(1/k)$ for convex smooth optimization. A breakthrough due to Nesterov [33] replaces the constant momentum $\gamma$ with $(k-1)/(k+2)$, a.k.a. NAG momentum, improves the convergence rate to $O(1/k^2)$, which is proved to be optimal for convex and smooth objective functions [33, 52]. We demonstrate the faster convergence of NAG in comparison with GD and HB on a quadratic optimization problem in Figure 1. The much faster convergence rate of NAG than that of GD and HB motivates us to incorporate the second-order ODE limit of NAG into a NODE and propose the NesterovNODE. Nesterov acceleration gradient method [33] takes the following form: given initial points $x_0 \in \mathbb{R}^N$ and $y_0 = x_0$, the sequence $\{(x_k, y_k)\}_k$ is defined inductively as:

$$\begin{cases} x_k = y_{k-1} - s\nabla F(y_{k-1}), \\ y_k = x_k + \dfrac{k-1}{k+2}(x_k - x_{k-1}). \end{cases} \tag{7}$$

The continuous limit of the Nesterov scheme is obtained by setting $x_k = \mathbf{h}(k\sqrt{s}) = \mathbf{h}(t)$ with $t = k\sqrt{s}$ and some smooth function $\mathbf{h}$ from $\mathbb{R}$ to $\mathbb{R}^N$. According to [52], the function $\mathbf{h}$ satisfies the Nesterov ODE

$$\mathbf{h}''(t) + \frac{3}{t}\mathbf{h}'(t) + \nabla F(\mathbf{h}(t)) = 0, \tag{8}$$

with the initial conditions $\mathbf{h}(0) = \mathbf{h}_0$, $\mathbf{h}'(0) = 0$.

**Remark 1** (Nesterov factor). *The constant $3$ in the coefficient of $\mathbf{h}'(t)$ in Eq. (8) is originally from the approximation $(k-1)/(k+2) = 1 - 3/k + \mathcal{O}(1/k^2)$. This constant will be replaced by a constant $r$ if the factor $(k-1)/(k+2)$ in Eq. (7) is replaced by $(k-1)/(k+r-1)$. It is proved in [52] that the Nesterov ODE still holds the quadratic convergence rate when $3$ is replaced by any number $r > 3$.*

**Remark 2** (Numerical stability). *If the Euler method is used, then to keep the numerical solution close to the exact solution, the step size chosen in the Euler method must be small enough. However, the smaller the step size, the more expensive the computation. The maximum stable step size, which is the maximum value for which the step size can be chosen so that the numerical solution remains close to the exact solution, reflects the numerical stability of the ODEs. It is proved in [52] that the maximum stable step size of the Nesterov ODE is much larger than that of the ODE (4), thus showing the numerical stability advantage of the Nesterov ODE.*

## 3 Generalize Nesterov ODEs to Differential-Algebraic Systems

One can parameterize $\nabla F(\mathbf{h}(t))$ in Eq. (8) by a neural network $f(\mathbf{h}(t), t, \theta)$ with learnable parameters $\theta$ in a similar way as NODE. This results in the following Nesterov Neural ODE (NesterovNODE)

$$\mathbf{h}''(t) + \frac{3}{t}\mathbf{h}'(t) + f(\mathbf{h}(t), t, \theta) = 0, \qquad (9)$$

This second-order NesterovNODE can be written in term of the first-order NesterovNODE as

$$\begin{cases} \mathbf{h}'(t) = \mathbf{m}(t), \\ \mathbf{m}'(t) = -\frac{3}{t}\mathbf{m}(t) - f(\mathbf{h}(t), t, \theta). \end{cases} \qquad (10)$$

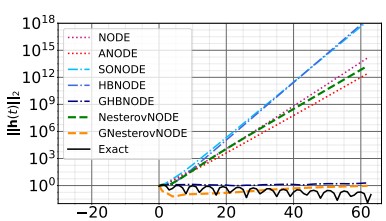

Figure 2: Contrasting the increase in the $l2$-norm of $h(t)$ for NODE, ANODE, SON-ODE, HBNODE, GHBNODE, NesterovN-ODE, and GNesterovNODE over long integration time on the Silverbox Initialization task (More details in Appendix D.1).

However, because of the singularity created by the coefficient $\frac{3}{t}$, the training process based directly on Eq. (9) and (10) will be unstable. To avoid the instability issue, we set $\mathbf{h}(t) = k(t)\mathbf{x}(t)$ with $k(t) = t^{-\frac{3}{2}}e^{\frac{t}{2}}$. Then Eq. (8) becomes

$$k(t)\mathbf{x}''(t) + \left(2k'(t) + \frac{3}{t}k(t)\right)\mathbf{x}'(t) + \left(k''(t) + \frac{3}{t}k'(t)\right)\mathbf{x}(t) + \nabla F(\mathbf{h}(t)) = 0. \qquad (11)$$

We observe that $2k'(t) + \frac{3}{t}k(t) = k(t)$. By dividing both sides of Eq. (11) by $k(t)$, we obtain:

$$\mathbf{x}''(t) + \mathbf{x}'(t) + f(\mathbf{h}(t), t) = 0, \qquad (12)$$

where

$$f(\mathbf{h}(t), t) = \frac{1}{4}\left(t^2 - 3\right)t^{-\frac{1}{2}}e^{-\frac{t}{2}}\mathbf{h}(t) + t^{\frac{3}{2}}e^{\frac{-t}{2}}\nabla F(\mathbf{h}(t)).$$

Let $\mathbf{m}(t) = \mathbf{x}'(t)$. Then Eq. (8) is equivalent to the following differential-algebraic system

$$\begin{cases} \mathbf{h}(t) = t^{\frac{-3}{2}}e^{\frac{t}{2}}\mathbf{x}(t), \\ \mathbf{x}'(t) = \mathbf{m}(t), \\ \mathbf{m}'(t) = -\mathbf{m}(t) - f(\mathbf{h}(t), t). \end{cases} \qquad (13)$$

We parameterize $f(\mathbf{h}(t), t)$ as a neural network, recalled as $f(\mathbf{h}(t), t, \theta)$ with learnable parameter $\theta$, and obtain the differential-algebraic version of NesterovNODE

$$\begin{cases} \mathbf{h}(t) = t^{\frac{-3}{2}}e^{\frac{t}{2}}\mathbf{x}(t), \\ \mathbf{x}'(t) = \mathbf{m}(t), \\ \mathbf{m}'(t) = -\mathbf{m}(t) - f(\mathbf{h}(t), t, \theta). \end{cases} \qquad (14)$$

The singularity at $t = 0$ of the NesterovNODE in Eqs. (9) and (10) is now moved to the algebraic equation of the above differential-algebraic system.

It is often the case when training ODE-based models that some functions diverge or explode at a finite time. This phenomenon is called blow-up, which we demonstrate in Fig. 2. In order to alleviate the blow-up problem, we introduce a generalized version of NesterovNODE, termed Generalized NesterovNODE (GNesterovNODE). For the NesterovNODE, the potential blow-up is due to the oscillation inherited from NAG scheme as can be seen in Fig. 1. The blow-up can also occur because of the singularity caused by the factor $t^{\frac{-3}{2}}e^{\frac{t}{2}}$ in the algebraic equation. Following the techniques presented in [60], we address these potential blow-up by applying an activation function $\sigma$ to the function $f(\mathbf{h}(t), t, \theta)$, the factor $t^{\frac{-3}{2}}e^{\frac{t}{2}}$, and the momentum state $\boldsymbol{m}(t)$ of the NesterovNODE. The activation function can be any activation function commonly used. In our experiments, we use $\texttt{tanh}$ and $\texttt{hardtanh}$. In addition, the residual term $\xi\mathbf{h}(t)$, which stands for a skip connection [16], is also added into the governing equation of $\boldsymbol{m}(t)$, which benefits training and generalization of GNesterovNODEs. The GNesterovNODE differential-algebraic system is then given by

$$\begin{cases} \mathbf{h}(t) = \sigma(t^{\frac{-3}{2}}e^{\frac{t}{2}})\mathbf{x}(t), \\ \mathbf{x}'(t) = \sigma(\mathbf{m}(t)), \\ \mathbf{m}'(t) = -\mathbf{m}(t) - \sigma(f(\mathbf{h}(t), t, \theta)) - \xi\mathbf{h}(t), \end{cases} \tag{15}$$

with the initial conditions $\mathbf{h}(0) = \mathbf{h}_0$, $\mathbf{x}(0) = \mathbf{x}_0$, $\mathbf{m}(0) = \mathbf{m}_0$. It is noted that, from $\mathbf{h}(0) = \mathbf{h}_0$ and $\mathbf{h}'(0) = 0$, we must have $\mathbf{x}_0 = \mathbf{h}_0 \lim_{t\to 0} \sigma(t^{\frac{-3}{2}}e^{\frac{t}{2}})^{-1}$ and $\mathbf{m}_0 = \mathbf{h}_0 \lim_{t\to 0} \frac{d}{dt}\sigma(t^{\frac{-3}{2}}e^{\frac{t}{2}})^{-1}$. Fig. 2 shows that GNesterovNODE can indeed control the growth of $\boldsymbol{h}(t)$ effectively.

In general, GNesterovNODEs form a wide class of neural differential equations which are represented by differential-algebraic systems and contain NesterovNODEs as a subclass. Eqs. (9) and (15) define the forward ODE for the (G)NesterovNODE. To efficiently update the parameters during the training process based on (G)NesterovNODEs, we use the continuous adjoint sensitivity given in Propositions 1 and 2 below.

**Proposition 1** (Continuous adjoint equation for the second-order NesterovNODE). *The continuous adjoint state* $\mathbf{a}(t) = \frac{\partial \mathcal{L}}{\partial \mathbf{h}(t)}$ *of the NesterovNODE given in Eq. (9) satisfies the following NesterovN-ODE*

$$\mathbf{a}''(t) - \frac{3}{t}\mathbf{a}'(t) + \mathbf{a}(t)\frac{\partial f(\mathbf{h}(t), t, \theta)}{\partial \mathbf{h}} + \frac{3}{t^2}\mathbf{a}(t) = 0. \tag{16}$$

**Proposition 2** (Continuous adjoint equation for GNesterovNODE). *The continuous adjoint state functions* $\mathbf{a_h}(t) := \frac{\partial \mathcal{L}}{\partial \mathbf{h}(t)}$, $\mathbf{a_x}(t) := \frac{\partial \mathcal{L}}{\partial \mathbf{x}(t)}$ *and* $\mathbf{a_m}(t) := \frac{\partial \mathcal{L}}{\partial \mathbf{m}(t)}$ *of the GNesterovNODE system given in Eq. (15) satisfy the following differential-algebraic adjoint system*

$$\begin{cases} \mathbf{a_h}(t) = \sigma(t^{\frac{-3}{2}}e^{\frac{t}{2}})^{-1}\mathbf{a_x}(t), \\ \mathbf{a_x}'(t) = t^{\frac{-3}{2}}e^{\frac{t}{2}}\mathbf{a_m}(t)\left[\frac{\partial \sigma(f(\mathbf{h}(t), t))}{\partial \mathbf{h}} + \xi\mathbf{I}\right], \\ \mathbf{a_m}'(t) = \mathbf{a_m}(t) - \mathbf{a_x}(t)\sigma'(\mathbf{m}(t)), \end{cases} \tag{17}$$

*with the final value conditions* $\mathbf{a_h}(T) = \frac{\partial L}{\partial \mathbf{h}_T}$, $\mathbf{a_x}(T) = \frac{\partial L}{\partial \mathbf{x}_T}$ *and* $\mathbf{a_m}(T) = \frac{\partial L}{\partial \mathbf{m}_T}$.

## 4    The Effectiveness of GNesterovNODE in Alleviating Vanishing Gradients

In neural networks with many layers, the vanishing gradient is one of the major issues [40]. In the cases of NODEs and their hybrid ODE-RNN variants, the vanishing gradient issue may occur when the adjoint state $\mathbf{a}(t) := \frac{\partial \mathcal{L}}{\partial \mathbf{h}(t)}$ goes to 0 quickly as $T - t$ increases. In this section, we will prove that this vanishing gradient issue can be avoided in GNesterovNODEs.

For the GNesterovNODE given in Eq. (15), the gradient $\frac{\partial \mathcal{L}}{\partial \mathbf{h}_t}$ can be determined from $\frac{\partial \mathcal{L}}{\partial \mathbf{x}_t}$ via the algebraic relation:

$$\frac{\partial \mathcal{L}}{\partial \mathbf{h}_t} = \frac{\partial \mathcal{L}}{\partial \mathbf{x}_t}\frac{\partial \mathbf{x}_t}{\partial \mathbf{h}_t} = \sigma(t^{\frac{-3}{2}}e^{\frac{t}{2}})^{-1}\frac{\partial \mathcal{L}}{\partial \mathbf{x}_t}.$$

While the gradients $\frac{\partial \mathcal{L}}{\partial \mathbf{x}_t}$ and $\frac{\partial \mathcal{L}}{\partial \mathbf{m}_t}$ satisfy the following proposition.

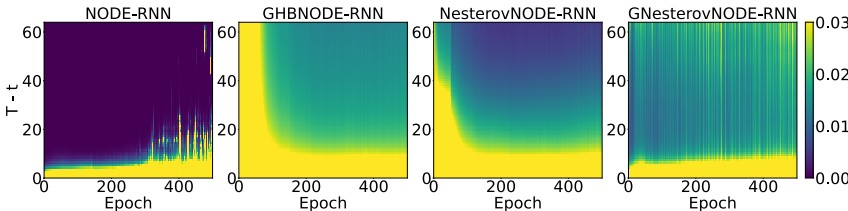

Figure 3: Plot of the $l2$-norm of the adjoint states for ODE-RNN, GHBNODE-RNN, NesterovNODE-RNN and GNesterovNODE-RNN back-propagated from the last time stamp. The term $T - t$ demonstrates the gap between the final time $T$ and intermediate time $t$. When the gap $T - t$ becomes larger, NesterovNODE-RNN and GNesterovNODE-RNN can address the vanishing gradient problem due to the adjoint states of these methods' decay slowly.

**Proposition 3.** *For every $t \in (0, T)$, there exist a unit length row vector $v \in \mathbb{C}^N$ and an upper triangular matrix $U \in \mathbb{C}^{N \times N}$ such that*

$$\left\| \begin{bmatrix} \frac{\partial \mathcal{L}}{\partial \mathbf{x}_t} & \frac{\partial \mathcal{L}}{\partial \mathbf{m}_t} \end{bmatrix} \right\|_2 = \| v \exp(U) \|_2 \left\| \begin{bmatrix} \frac{\partial \mathcal{L}}{\partial \mathbf{x}_T} & \frac{\partial \mathcal{L}}{\partial \mathbf{m}_T} \end{bmatrix} \right\|_2,$$

*and at least $\frac{N}{2}$ complex values in the diagonal of $U$ have the real parts greater than or equal to $\frac{t-T}{2}$.*

The term $v \exp(U)$ in the above proposition plays the essential role in the nonvanishing gradient issue. Without loss of generality, we can assume that $U = \begin{bmatrix} U_{\text{large}} & P \\ \mathbf{0} & U_{\text{small}} \end{bmatrix}$ where all complex numbers in the diagonal of $U_{\text{large}}$ (resp. $U_{\text{small}}$) have the real parts greater than or equal to (resp. smaller than) $\frac{1}{2}(t - T)$. According to Proposition 3, the size of $U_{\text{large}}$ is at least $N/2$. Then we have,

$$\exp(U) = \begin{bmatrix} \exp(U_{\text{large}}) & \tilde{P} \\ \mathbf{0} & \exp(U_{\text{small}}) \end{bmatrix} \quad \text{and} \quad \| v \exp(U) \|_2 \geq \| v_{\text{large}} \exp(U_{\text{large}}) \|_2.$$

Here, the vector $v_{\text{large}}$ is the first $m$ columns of $v$, and $m$ is the size of $U_{\text{large}}$. Since the real parts of elements in the diagonal of $U_{\text{large}}$ is no less than $\frac{1}{2}(t - T)$, $\exp(U_{\text{large}})$ decays at a rate at most $\frac{1}{2}(t - T)$. This results in the nonvanishing gradient of $\begin{bmatrix} \frac{\partial \mathcal{L}}{\partial \mathbf{x}_t} & \frac{\partial \mathcal{L}}{\partial \mathbf{m}_t} \end{bmatrix}$, hence so is $\begin{bmatrix} \frac{\partial \mathcal{L}}{\partial \mathbf{h}_t} & \frac{\partial \mathcal{L}}{\partial \mathbf{x}_t} & \frac{\partial \mathcal{L}}{\partial \mathbf{m}_t} \end{bmatrix}$.

To illustrate, we take the Walker2D kinematic simulation task [2] in consideration, which requires learning long-term dependency effectively [27]. We train ODE-RNN [46], GHBNODE-RNN [60], NesterovNODE-RNN and GNesterovNODE-RNN on this benchmark dataset (More details in Appendix D.4). Fig. 3 plots $\left\| \frac{\partial \mathcal{L}}{\partial \mathbf{h}_t} \right\|_2$ for ODE-RNN, $\left\| \begin{bmatrix} \frac{\partial \mathcal{L}}{\partial \mathbf{h}_t} & \frac{\partial \mathcal{L}}{\partial \mathbf{m}_t} \end{bmatrix} \right\|_2$ for GHBNODE-RNN and $\left\| \begin{bmatrix} \frac{\partial \mathcal{L}}{\partial \mathbf{h}_t} & \frac{\partial \mathcal{L}}{\partial \mathbf{x}_t} & \frac{\partial \mathcal{L}}{\partial \mathbf{m}_t} \end{bmatrix} \right\|_2$ for NesterovNODE-RNN and GNesterovNODE-RNN, showing that when the gap between the final time $T$ and intermediate time $t$ becomes larger, the adjoint states of NesterovNODE-RNN, GNesterovNODE-RNN and GHBNODE-RNN decay much more slowly than NODE-RNN. Thus, NesterovNODE-RNN and GNesterovNODE-RNN have the ability to tackle the vanishing gradient issue.

**Remark 3.** *The gradient exploding problem can be effectively resolved via gradient clipping, training loss regularization, etc. [40, 10]. Therefore, in practice the vanishing gradient problem is the major issue for training deep neural networks [40].*

## 5 Experimental Results

In this section, we empirically study the advantages of our proposed NesterovNODE/GNesterovN-ODE over the baseline NODEs and other popular NODE-based architectures, including the augmented NODE (ANODE) [9], the Second Order NODE (SONODE) [37], HBNODE/GHBNODE [60] on a variety of benchmarks including point cloud separation, image classification, and kinetic simulation which involve different data modalities ranging from point cloud to images and time series. ANODEs augments the space on which the ODE is solved while SONODEs and (G)HBNODEs solve a second-order ODE. We aim to show that: (i) NesterovNODEs/GNesterovNODEs require significantly fewer NFEs while attaining similar or even better accuracy as the baselines; (ii) GNesterovNODEs avoid the blow-up of $\mathbf{h}(t)$ and thus improve over NesterovNODEs; (iii) NesterovNODEs/GNesterovNODEs

Table 1: The parameters count for the models in point cloud separation, image classifications, and the Walker2D kinematic simulation tasks and the test accuracy on CIFAR10/MNIST. Our methods are able to reach similar or better test accuracy than the baseline methods on CIFAR10 while retaining a similar test accuracy on MNIST.

| Model | Number of parameters | | | | Test Accuracy | |
|---|---|---|---|---|---|---|
| | PC | MNIST | CIFAR10 | Walker2D | CIFAR10 | MNIST |
| NODE | 545 | 85316 | 173611 | 9929 | $0.5466 \pm 0.0051$ | $0.9531 \pm 0.0042$ |
| ANODE | 587 | 85462 | 172452 | 10019 | $0.6025 \pm 0.0032$ | $0.9816 \pm 0.0024$ |
| SONODE | 541 | 86179 | 171635 | 11471 | $0.6132 \pm 0.0073$ | $\mathbf{0.9824 \pm 0.0013}$ |
| HBNODE | 582 | 85931 | 172916 | 10099 | $0.5989 \pm 0.0035$ | $0.9814 \pm 0.0011$ |
| GHBNODE | 582 | 85931 | 172916 | 10099 | $0.6085 \pm 0.0050$ | $0.9817 \pm 0.0005$ |
| NesterovNODE | 581 | 85930 | 172915 | 10098 | $0.5996 \pm 0.0033$ | $\mathbf{0.9824 \pm 0.0015}$ |
| GNesterovNODE | 581 | 85930 | 172915 | 10098 | $\mathbf{0.6172 \pm 0.0064}$ | $0.9807 \pm 0.0013$ |

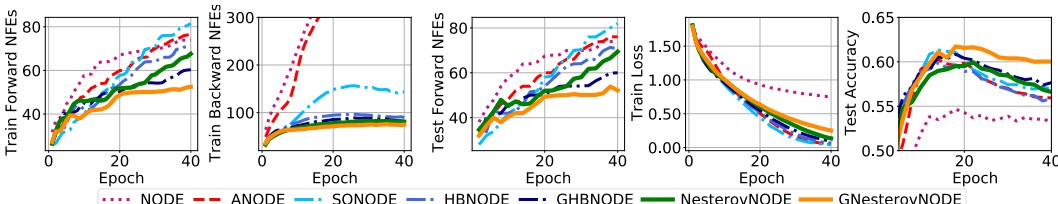

Figure 4: Contrasting the NFEs and accuracy of NODE-based baselines and our methods NesterovNODE/GNesterovNODE on the CIFAR10 dataset (Tolerance: $10^{-5}$).

capture better long-term dependencies than the baselines and achieve better results in long-sequence modeling tasks.

For all the experiments, we use Adam [23] as the optimizer and Dormand-Prince 5(4) [7] as the numerical ODE solver. We choose the network architecture used to parameterize $f(\boldsymbol{h}(t), t, \theta)$ so that our proposed models and the baselines have similar numbers of parameters in our experiments as shown in Table 1. Other training/model/dataset details are provided in Appendix D. All results are averaged over 5 runs with different seeds. We conduct the experiments on a server with 6 NVIDIA 2080Ti GPUs with 11GB of GPU memory. Our PyTorch code with documentation can be found at https://github.com/minhtannguyen/NesterovNODE.

## 5.1 Image classification

We validate the accuracy and efficiency advantage of NesterovNODE/GNesterovNODE for image classification on MNIST [6] and CIFAR10 [25] in comparison with other ODE-based baselines. We follow the same training and model settings as in [60].

**NFEs.** As shown in Fig. 4, our NesterovNODE and GNesterovNODE reduce the NFEs in both the forward and the backward propagations compared to the baseline models. Although the augmented input dimensions in ANODE help reduce the NFEs compared to NODE, second-order methods reduce the NFEs more significantly. Compared to the other second-order methods i.e. SONODE, HBNODE, and GHBNODE, GNesterovNODE achieve much better NFE reductions on both MNIST (see Fig 12 in the Appendix) and CIFAR10, indicating the improvement in efficiency and stability of our methods over the other second-order baseline models. Such advancement is an essential step to scale GNesterovNODE to larger and more complex practical tasks.

**Accuracy.** Table 1 shows that our GNesterovNODE achieves the highest test accuracy on CIFAR10. On MNIST, NesterovNODE attains the second-highest test accuracy and very close to the best result from SONODE while being much more efficient than SONODE. This advantage in terms of accuracy can be associated with the small number of NFEs needed by NesterovNODE and GNesterovNODE above, which reduces the model complexity and leads to better generalization.

Note that GNesterovNODE improves over NesterovNODE in efficiency and accuracy (on larger and more challenging benchmark like CIFAR10). This justifies the effectiveness of our solution that introduces an additional bounded activation function $\sigma$ and the residual term $\xi \boldsymbol{h}(t)$ to prevent the blow-up of $\boldsymbol{h}(t)$ as explained in Section 3.

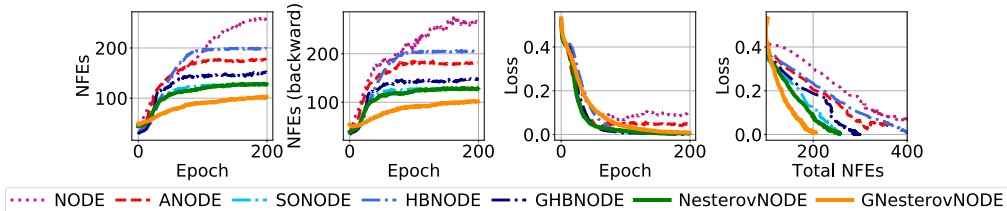

Figure 5: Contrasting the NFEs and training loss of NODE-based baselines and our NesterovNODE/GNesterovNODE on the point cloud benchmark. Results are averaged over $50$ runs (Tolerance: $10^{-7}$).

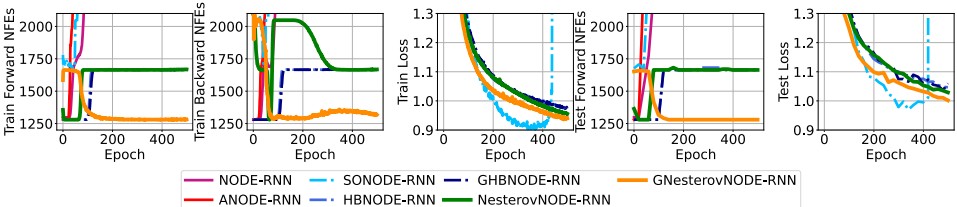

Figure 6: Contrasting the NFEs and losses of NODE-RNN [4], ANODE-RNN [9], HBNODE-RNN/GHBNODE-RNN [60], and our NesterovNODE-RNN/GNesterovNODE-RNN on the Walker2D dataset (Tolerance: $10^{-7}$).

## 5.2 Point cloud separation

We perform experiments on a point cloud separation task in order to verify that NesterovNODEs/GNesterovNODEs can learn effective features to separate two sets of point clouds. The first set consists of $40$ points drawn from a circle with the radius $||\boldsymbol{r}|| < 0.5$, while the second set comprises $80$ points drawn from an annulus with the inner and outer radius of $0.85$ and $1$, respectively, i.e. $0.85 < ||\boldsymbol{r}|| < 1.0$. Fig. 5 shows that our NesterovNODE and GNesterovNODE models are able to converge to 0 loss consistently while other methods have difficulty to reach 0 loss. In addition, NesterovNODE and especially the GNesterovNODE require significantly fewer NFEs in forward and backward passes compared to the baselines. Thus, NesterovNODE and GNesterovNODE help improve both the training and the efficiency of the model. We plot the evolution of the point cloud separation through 100 epochs for a random run of each model in Appendix D.2. Like SONODE, HBNODE/GHBNODE, NesterovNODE/GNesterovNODE learn effective features that allow good separation between the two classes of point clouds in these experiments while NODE and ANODE fail for this task.

## 5.3 Walker2D kinematic simulation

In this section, we investigate NesterovNODE/GNesterovNODE when applied on time-series data. In particular, we use the ODE-RNN framework [46], with the recognition model being set to different ODE-based models, to study Walker2D kinematic simulation task, which requires learning long-term dependency effectively [26]. As shown in Fig. 6, our NesterovNODE-RNN and GNesterovNODE-RNN not only reduce the NFEs both in the forward and the backward stages, but also achieve smaller test loss compared to the baseline models that we compare with, including (G)HBNODE-RNN, ANODE-RNN, and NODE-RNN. Although SONODE-RNN achieves much smaller losses, its NFEs are too high, thus training SONODE-RNN for this task is much more time-consuming compared to our Nesterov-based methods.

## 5.4 Continuous Normalizing Flows for MNIST

We compare the GNesterovNODE with the baseline NODE and GHBNODE for use in variational inference with the continuous normalizing flow model trained on the MNIST dataset. The continuous normalizing flow model we use is the FFJORD in [14]. We summarize our results in Figure 7. Compared to the FFJORD-NODE and FFJORD-GHBNODE, the FFJORD-GNesterovNODE significantly reduces the NFEs in both forward and backward passes while improving the negative ELBO on the test set. This experiment demonstrates that our method GNesterovNODE accelerates NODE-based models on both discriminative tasks and generative tasks.

## 6 Observed Properties of NesterovNODE in GNesterovNODE

In this section, we verify empirically that with the addition of the activation function $\sigma$ and the skip connection, GNesterovNODEs still preserve important properties of NesterovNODEs as stated in Re-

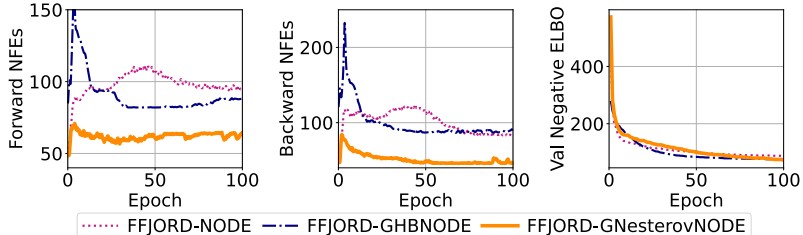

Figure 7: Contrasting the NFEs and the validation negative ELBO of the FFJORD-NODE, the FFJORD-HBNODE, and our FFJORD-GNesterovNODE for the variational inference task with a continuous normalizing flow model, i.e. FFJORD [14], on the binarized MNIST dataset (Tolerance: $10^{-5}$).

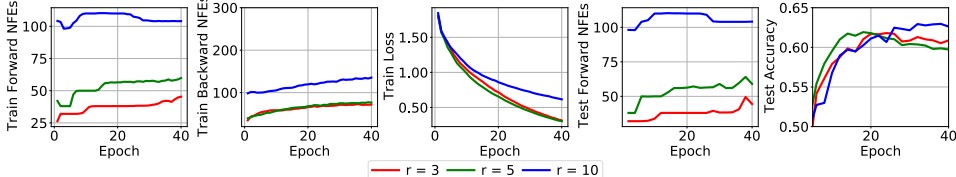

Figure 8: Contrasting different values of the factor $r$ on CIFAR10 with GNesterovNODE (Tolerance: $10^{-5}$).

mark 1 and Remark 2. This hints that GNesterovNODEs have the same behaviors as NesterovNODEs while enjoying more stable training.

**Effect of the Nesterov factor on NesterovNODE (Remark 1)**

As stated in [52], the "magic constant" 3 can be replaced by a constant $r > 3$ while maintaining a convergence rate of $O(1/k^2)$. In this work's experiments, a larger $r$ leads to a better test loss eventually despite a smaller $r$ outperforming in the beginning, but their experiments are based on Nesterov ODE. This section investigates whether this behavior extends to our generalized method GNesterovNODE. As shown in Fig. 8 and Table 2, a larger $r$ also leads to a higher test accuracy, and for $r = 3$ and $r = 5$, the model converges to its best test accuracy sooner than $r = 10$. Although the training loss of $r = 10$ is larger than $r = 3$ and $r = 5$, we observe that all three values of $r$ reach their best testing accuracy when the training loss is approximately in the range of $[0.65, 0.75]$, which hints that decreasing the loss further leads to overfitting. One more interesting observation is that a higher value of $r$ increases the forward and backward NFEs. Intuitively speaking, the term $-\frac{r}{t}\mathbf{h}'(t)$ in the NesterovNODE, which is opposite to the product of the damping parameter $r/t$ and the velocity $\mathbf{h}'(t)$, represents the friction force of the model (see [52, Section 4]). When $r$ is larger, the friction force resists the movement of $\mathbf{h}(t)$ along the trajectory stronger, which slows down the convergence of training loss.

**Stability of NesterovNODE (Remark 2)**

The NesterovNODE is more numerically stable than NODE in the sense that the step size in the Euler method for solving the NesterovNODE can be chosen larger while the stability of the numerical solution is still guaranteed (see Remark 2). We hypothesize that this fact still holds for the GNesterovNODE in comparison with both NODE and GHBNODE. To illustrate this fact, we perform experiments on CIFAR10 using Euler solvers with large step sizes $(0.1, 0.2, 0.5)$. Due to the instability of large step size, GHBNODE moves fast to the maximum accuracy points and then goes down, as shown in Fig. 9, while NODE achieves high train loss and low test accuracy. Even when the step size is big, GNesterovNODE's training is stable as the curve evolves gradually. More interestingly, as shown in Fig. 10, GNesterovNODE solved with rk4 (Fourth-order Runge-Kutta with 3/8 rule) solver using a large step size outperforms the same model solved with the adaptive step size solvers like dopri5 (Dormand-Prince of order 5) solver. This show the promise of using large step sizes in GNesterovNODE to obtain models with better accuracy and efficiency.

Table 2: Test accuracy for GNesterovNODE on CIFAR10 with varying Nesterov factors $r$ (Tolerance: $10^{-5}$).

| Value of $r$ | Test accuracy |
|:---:|:---:|
| 3 | 0.6180 |
| 5 | 0.6192 |
| 10 | 0.6296 |

## 7 Related Work

**Increasing efficiency of training NODEs.** Several methods have been proposed to reduce the NFEs in NODEs and increase the model efficiency. Among them are works that use weight decay [14] and

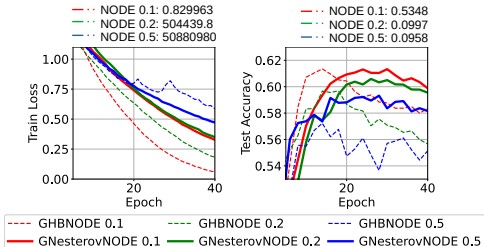

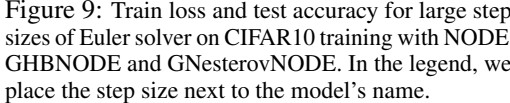

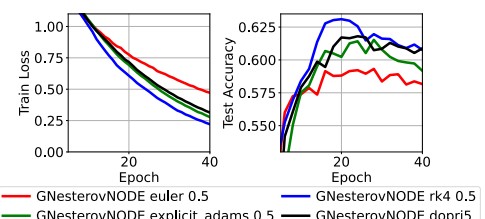

Figure 9: Train loss and test accuracy for large step sizes of Euler solver on CIFAR10 training with NODE, GHBNODE and GNesterovNODE. In the legend, we place the step size next to the model's name.

Figure 10: Train loss and test accuracy for various solvers with large step size $0.5$ on CIFAR10 training with GNesterovNODE. In the legend, we place the step size next to the model's name.

other regularizers applied on the solver and the learned dynamics [11, 20, 12, 39]. Other works employ input augmentation [8], data-control [30] and depth-variance [30, 36] to reduce the NFEs needed to compute the ODE solution. Another method replaces norms with seminorms in the backward continous adjoint stage to reduce backward NFEs [22]. NesterovNODE solves second-order Nesterov ODE to reduce both forward and backward NFEs.

**Second-order dynamical systems.** Second-order ODEs have also been employed to speed up NODEs. SONODE [37] replaces the first-order ODE in Eq. (1) by a second ODE which can be solved as a system of first-order ODEs. HBNODE [60] also solves a second-order ODE but with an additional constant damping parameter, which corresponds to the ODE limit of the HB momentum method. NesterovNODE solves the second-order Nesterov ODE with a time-dependent damping parameter, which corresponds to the ODE limit of the NAG momentum method. These momentum-based systems have also been employed in designing deep neural networks as in [32, 28, 49, 35].

**Learning long-term dependencies.** The ability of a model to learn long-term dependencies is highly needed to scale up the model to large-scale tasks that involve very long sequences. Existing works try to alleviate exploding or vanishing gradient issues happened during the training of recurrent neural networks, including [1, 59, 19, 55, 31, 17, 34]. Recently, learning long-term dependencies with NODEs has been explored. For example, [26] integrate a long-short term memory cell into NODEs. Among the hallmarks of NesterovNODEs is that our proposed models can directly capture long-term dependencies in long sequences.

# 8 Concluding Remarks

In this paper, we propose the NesterovNODE and its generalized version GNesterovNODE that solve the second-order ODE limit of NAG. These models take advantage of the convergence rate $\mathcal{O}(1/k^2)$ of the NAG scheme to gain acceleration over NODEs and the existing NODE-based models such as HBNODE by reducing the NFEs in solving both forward and backward ODEs. Our Nesterov-based NODEs also achieve better accuracy than NODEs and outperform or at least are on par with other NODE-based models in our experiments while requiring much fewer NFEs. We also prove that NesterovNODEs and GNesterovNODEs can avoid the vanishing gradient issue and can capture long-term dependencies in long sequences effectively. It is worth mentioning that NesterovNODEs/GNesterovNODEs do not introduce any inherently negative societal impact. The high-resolution ODE in [51, 50] is an alternative way to take a continuous-time limit of the NAG scheme and heavy-ball momentum method. This high-resolution ODE introduces a gradient correction that allows the NAG scheme to achieve an inverse cubic rate for minimizing the squared gradient norm, which is better than the inverse square rate in the low-resolution ODE that our method uses. A limitation of our paper is that we have not incorporated restart schemes [38, 45, 56] into the NesterovNODE, and we leave this as future work.

## Acknowledgements

This material is based on research sponsored by the AFOSR MURI FA9550-18-1-0502, the ONR grant N00014-20-1-2093, the MURI N00014-20-1-2787, and the NSF under Grant# 2030859 to the Computing Research Association for the CIFellows Project (CIF2020-UCLA-38). NH acknowledges support from the NSF IFML 2019844 and the NSF AI Institute for Foundations of Machine Learning.

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
