# OpenReview forum: "Improving Neural Ordinary Differential Equations with Nesterov's Accelerated Gradient Method"
_NeurIPS.cc/2022/Conference — NeurIPS 2022 Accept_

### Official Review · Reviewer_hkAe · 2022-07-08

**Rating:** 7
**Confidence:** 4
**Soundness:** 4 excellent
**Presentation:** 4 excellent
**Contribution:** 2 fair

**Summary:**

This paper takes Nesterov's Accelerated Gradient Method, and applies it to Neural ODEs. One can think of a Neural ODE as carrying out vanilla gradient descent where:

$\frac{dh}{dt} = -\nabla_{h}F(h, t)$

Heavy ball Neural ODEs carry out the continuous limit of heavy ball momentum accelerated gradient descent where:

$\frac{d^2h}{dt^2} = -\gamma\frac{dh}{dt} - \nabla_{h}F(h, t)$

Finally, this work takes the continuous limit of Nestervov accelerated gradient descent:

$\frac{d^2h}{dt^2} = -\frac{3}{t}\frac{dh}{dt} - \nabla_{h}F(h, t)$

The work demonstrates that this style of second order differential equation converges to its equilibrium point significantly faster than heavy ball Neural ODEs and other variations of Neural ODE. In turn this requires far fewer function evaluations in the ODE solve, while maintaining performance.

Proofs are provided to show that the method will be resistant to vanishing gradients and the solution will require fewer function evaluations. Extensive experiments are done on a large collection of Neural ODE variants, demonstrating performance is just as good but the speed of the ODE solve is greater.

**Questions:**

- Does the Nesterov method only really make sense for end point style tasks? That is, ones where the trajectory doesn't matter, so not time-series.
- Where does the oscillation in Figure 1 come from?
- Remark 1 talks about $\frac{k-1}{k-3}$ but in the previous paragraph and equation 7 we have $\frac{k-1}{k+2}$. Which one is it? Does this affect any results?
- Is applying the sigmoid function to stabilize Nesterov theoretically motivated, or more of an empirical trick to stabilize the ODE solve?
- The analysis of vanishing gradients is great, however in the worst case it still seems that we can get $\exp(-T)$ by the time we reach the start of the solve. It seems from the experiments this doesn't happen much. Has a similar analysis been done on other Neural ODE variants? To provide theory which supports the experiment in Figure 3
- Line 142, by writing $h(t)$ as $k(t)x(t)$, the singularity has moved from the ODE to $k(t)$, how has this actually helped, without the sigmoid being used the singularity is still there, just moved to $k(t)$
- Remark 2 talks about the Euler method to solve the ODE and Nesterov ODE. Is there a statement that can be made for adaptive step solvers? For examples a larger tolerance can be used for Nesterov over ODE?
- Has any timing taken place, to see the wall clock advantage of Nesterov?

**Limitations:**

The paper addresses both the limitations and the negative societal impact briefly in the conclusion.

**Strengths And Weaknesses:**

Strengths:

The paper is very well written and well motivated. The experiments test the claims well. The theoretical findings are strong and show Nesterov and GNesterov are useful to reduce the number of NFE. The ablations are also very good.

Weaknesses:

There aren't many weaknesses to this paper, it is overall quite simple which is a good thing, but overall means the contribution is incremental and not necessarily ground breaking. In my opinion this is not a bad thing. Specific weaknesses:

- The experiments are conducted on problems that Neural ODEs aren't typically used for, (classification and point clouds). It would be more convincing to demonstrate this on continuous normalizing flow tasks.
- The other main application of Neural ODEs is to time-series, and Nesterov does worse than SONODE, likely due to restricting the dynamics
- The proposed method feels quite simple (seems like a theoretically motivated adaptation of Heavy Ball NODE)

Minor Mistakes/points (do not affect quality of paper):

- I find Generalized Nesterov to be a misleading term, I believe something like Stabilized Nesterov is more accurate, as a sigmoid is applied to part of the Nesterov method, rather than generalizing, it is more like adapting
- On the MNIST task, GNesterov is not second best, Nesterov is. GNesterov appears to be second worst, just after Neural ODE (line 224 and table 1)
- On the point cloud task, in my experience only vanilla Neural ODE has problem separating the clouds, all others are able to. The experiment has also been called $g$ in the ANODE paper and Nested Spheres in Dissecting Neural ODEs
- Appendix line 515/eq 19: Has $a(T) = -I$ it should be $a(T) = \frac{\partial L}{\partial z(T)}$

Minor Typos (do not affect quality of paper):

- Line 25: "in neural network" should be "in a neural network"
- Equations in the paper should ideally be in brackets, these can be manually put in, or done using eqref in latex
- Line 83: It would make more sense to use $x_{k} = x_{k-1}...$ than $x_{t} = x_{t-1}...$ because $k$ is used in later equations for discrete changes (equation 7 for example)
- Line 200 and 318: "much fewer NFE" should be changed for "far fewer NFE" or "significantly fewer NFE" or simply "fewer NFE"
- $\mathcal{O}(k^2)$ and $\mathcal{O}(1/k^2)$ are used interchangeably in the paper to describe the convergence of Nesterov, it would be better to stay consistent
- Appendix footnote 1: "ANODDE" should be "ANODE"

---

> ### Author Response · Authors · 2022-08-02
> **Response to Review hkAe (1)**
>
> Thank you for your thoughtful review and valuable feedback. Below we address your concerns.
>
> ---
>
> **Q1. The experiments are conducted on problems that Neural ODEs aren't typically used for, (classification and point clouds). It would be more convincing to demonstrate this on continuous normalizing flow tasks.**
>
> **Reply:** Our NesterovNODE has two main advantages over other NeuralODEs model:
>
> 1. NesterovNODE helps reduce the number of NFEs needed for solving the ODEs in both the forward and backward passes thanks to the quadratic convergence rate of the Nesterov’s scheme.
>
> 2. NesterovNODE helps alleviate the vanishing gradient issue during training thanks to its well-structured spectrum.
>
> The experiments conducted in our paper focus on justifying these two advantages of NesterovNODE. Even though NeuralODEs are not typically used for image classification and point cloud tasks, these tasks can still serve as testbeds to study the behavior and efficiency of a new NeuralODE model. We are currently applying NesterovNODE to other applications including the continuous normalizing flow tasks as the reviewers suggest and will report those results in the discussion stage.
>
>
> **Q2. The other main application of Neural ODEs is to time-series, and Nesterov does worse than SONODE, likely due to restricting the dynamics**
>
> **Reply:** In the time-series experiment, even though the SONODE obtains better train and test loss, it is much more computationally expensive than NesterovNODE and GNesterovNODE.
>
> **Q3. The proposed method feels quite simple (seems like a theoretically motivated adaptation of Heavy Ball NODE)**
>
> **Reply:**  To derive the GNesterovNODE, and then GNesterovNODE,  is not simple. Please allow us to explain this by clarifying the key contributions needed to derive the (G)NesterovNODE.
>
> First, we need to use an appropriate change of variables to turn the NesterovODE with a time-dependent damping coefficient $3/t$ into a second-order ODE with a constant damping coefficient that is easier and more efficient to solve.
>
> Second, we need to convert the NesterovODE into its differential-algebraic version to avoid the singularity at $t=0$. As pointed out by Reviewer Uc87, “"neural DAEs" have received essentially no attention whatsoever, so this is a very interesting line of work to pursue.”
>
> Third, we prove that the adjoint equation for the second-order Nesterov NODE is also a second-order Nesterov NODE in Proposition 1. This result is more difficult to derive than the adjoint equation of the Heavy-Ball NODE in [Xia et al.(2021)].
>
> [1] Xia, H., Suliafu, V., Ji, H., Nguyen, T. M., Bertozzi, A., Osher, S., and Wang, B. Heavy ball neural ordinary differential equations. In Advances in Neural Information Processing Systems, 2021.
>
> **Q4. I find Generalized Nesterov to be a misleading term, I believe something like Stabilized Nesterov is more accurate, as a sigmoid is applied to part of the Nesterov method, rather than generalizing, it is more like adapting**
>
> **Reply:** We agree with the reviewer that “stabilized” or “modified” can be a better word. For our current revision, in order to avoid confusion during the rebuttal and discussion period, we still use “generalized” in the sense that the GNesterovNODE will become the NesterovNODE when in Eq. 15 in our paper, $\phi(\cdot)$ is an identity function and the coefficient $\xi$ of the skip connections is 0.
>
> **Q5. On the MNIST task, GNesterov is not second best, Nesterov is. GNesterov appears to be second worst, just after Neural ODE (line 224 and table 1)**
>
> **Reply:** Thanks for pointing this out. This is a typo. We meant NesterovNODE is the second best. We have fixed this typo in our revision.
>
> **Q6. On the point cloud task, in my experience only vanilla Neural ODE has problem separating the clouds, all others are able to. The experiment has also been called $g$ in the ANODE paper and Nested Spheres in Dissecting Neural ODEs**
>
> **Reply:** We agree with the reviewer that on this point cloud task, most methods can separate the cloud except the vanilla NeuralODE. In this experiment, we plan to show two things: 1) the NesterovNODE and GNesterovNODE can separate the cloud well like other methods, and 2) The NesterovNODE and GNesterovNODE are more efficient than other methods, requiring much fewer NFEs compared to the others.
>
> **Q7. Appendix line 515/eq 19: Has $a(T)=−I$  it should be $a(T) = dL/dz(T)$. Line 25: "in neural network" should be "in a neural network" ... "should be ANODE"**
>
> **Reply:** Thanks for pointing this out. We have addressed these points in the revision.

---

> > ### Author Response · Authors · 2022-08-02
> > **Response to Reviewer hkAe (2)**
> >
> > **Q8. Does the Nesterov method only really make sense for end point style tasks? That is, ones where the trajectory doesn't matter, so not time-series.**
> >
> > **Reply:** The continuous limit of NAG is also an ODE. When replacing the gradient term in Eq. 5 in our paper by a neural network, we turn this ODE limit into a second-order NODE, the NesterovNODE in Eq. 6, which like other NODEs, can fit the hidden dynamics in the latent space to learn relevant features by learning the parameters of the neural network $f(x(t), t, \theta)$ in Eq. 6 from the training data. Thus, NesterovNODE, like other NODEs, are good models for time-series and other tasks in which the trajectory matters. In the meantime, compared to the baseline first-order NODE, the NesterovNODE converges faster, reduces the NFEs, and thus improves the model’s efficiency.
> >
> > **Q9. Where does the oscillation in Figure 1 come from?**
> >
> > **Reply:** Oscillations along the trajectory of iterates approaching the minimizer are often observed when running Nesterov’s scheme. As suggested in [Su et al.(2014)] Using the ODE limits of Nesterov given by Eq. 8 in our paper, we can interpret this oscillation as follows.
> > When $t$ is small, in the beginning, the damping ratio $3/t$ is large. This leads the ODE to be an overdamped system, returning to the equilibrium without oscillating.
> > When $t$ is large $t$, as $t$ increases, the ODE with a small $3/t$ behaves like an underdamped system, oscillating with the amplitude gradually decreasing to zero.
> >
> > [1] Su, W., Boyd, S., and Candes, E. A differential equation for modeling nesterov’s accelerated gradient method: Theory and insights. In Advances in Neural Information Processing Systems, pp. 450 2510–2518, 2014.
> >
> > **Q10. Remark 1 talks about k−1/k−3 but in the previous paragraph and equation 7 we have k−1/k+2. Which one is it? Does this affect any results?**
> >
> > **Reply:** Thanks for pointing this out. $k-1/k-3$ in Remark 1 is a typo. It should be $k-1/k+2$. Other results in Remark 1 still hold.
> >
> > **Q11. Is applying the sigmoid function to stabilize Nesterov theoretically motivated, or more of an empirical trick to stabilize the ODE solve?**
> >
> > **Reply:** From our empirical study given in Figure 2,  we observe that the hidden state $h(t)$ of ANODEs, SONODEs, HBNODEs, and NesterovNODEs usually grows much faster than that of NODEs. The fast growth of $h(t)$ can lead to finite-time blow-up. Thus we propose to add the skip connection and apply the sigmoid function to the function $f(h(t), t, \theta)$, the factor $t^{-3/2}e^{t/2}$, and the momentum state $m(t)$ in Nesterov to mitigate this potential blow-up problem. It is an empirical approach to stabilize the NesterovNODE.
> >
> > **Q12: The analysis of vanishing gradients is great, however in the worst case it still seems that we can get exp⁡(−T) by the time we reach the start of the solver. It seems from the experiments this doesn't happen much. Has a similar analysis been done on other Neural ODE variants? To provide theory which supports the experiment in Figure 3**
> >
> > **Reply:** The adjoint state decays at a rate at most $exp((t-T)/2)$. Therefore, although we can get $exp(-T/2)$ by the time when we reach the start of the solver, this rate does help to avoid the vanishing gradient issue. A similar analysis can be considered for NODE and ANODE by using Eq.(23) in Appendix C. However, all eigenvalues of the matrix given by the integration in Eq.(23) can be arbitrarily large which results in the vanishing gradients. In contrast, a similar analysis in [Xia et al., 2021] shows that, like GNesterovNODE, the GHBNODE can help to avoid the vanishing gradients.
> >
> > **Q13. Line 142, by writing h(t) as k(t)x(t), the singularity has moved from the ODE to k(t), how has this actually helped, without the sigmoid being used the singularity is still there, just moved to k(t)**
> >
> > **Reply:** Moving the singularity to $k(t)$ in the algebraic equation, therefore removing the singularity out of the ODE solvers, does help to have stable numerical results when solving the last two differential equations in the DAE (15) and updating $x(t)$. The singularity in $k(t)$ does not affect much when updating $h(t)$ from $x(t)$.

---

> > > ### Author Response · Authors · 2022-08-03
> > > **Response to Reviewer hkAe (3)**
> > >
> > > **Q14. Remark 2 talks about the Euler method to solve the ODE and Nesterov ODE. Is there a statement that can be made for adaptive step solvers? For examples a larger tolerance can be used for Nesterov over ODE?**
> > >
> > > **Reply:** We have trained CIFAR10 with NODE, GHBNODE and GNesterovNODE using larger tolerance ($10^{-4}$) of dopri5 adaptive solver. The experimental result shows that GNesterovNODE is more stable and better at accuracy comparing to NODE and GHBNODE despite of large tolerance of adaptive solver.
> > >
> > > **Q15. Has any timing taken place, to see the wall clock advantage of Nesterov?**
> > >
> > > **Reply:** Thank you for the suggestion. We have provided the wall-clock time comparison between (G)NesterovNODEs and baseline models in Appendix E.4 of our revision.
> > >
> > > ---
> > > We hope we have cleared your concerns about our work. We have also revised our manuscript according to your comments, and we would appreciate it if we can get your further feedback at your earliest convenience.

---

> ### Comment · Reviewer_hkAe · 2022-08-03
> **Thank you for your answers - CNFs and Time-Series would be more convincing**
>
> Thank you to the authors for their responses. The answers have mostly answered all questions I have.
>
> If later in the week there are results on some continuous normalizing flows or times-series regression then I will raise my score. I still think with the added constraints of a Nesterov ODE it may be difficult to learn a time-series, because there is damping in the ODE. Correct me if I'm wrong, but  Nesterov ODEs may struggle to learn something like an orbiting planet where there is no damping present?
>
> For now I will keep my score the same, I think this paper presents an interesting line of research and the contributions are solid. As I say I would consider increasing if those experiments (which I believe ultimately will make the paper more convincing) on CNFs and time-series are done.
>
> Example CNFs can be found here: FFJORD https://arxiv.org/abs/1810.01367 (Grathwohl et al., 2018)
> Some simple time-series datasets are here: https://machinelearningmastery.com/time-series-datasets-for-machine-learning/

---

> > ### Author Response · Authors · 2022-08-04
> > **Thanks for your endorsement!**
> >
> > Thanks for your further feedback and we appreciate your endorsement. We are running additional experiments to verify the advantage of our NesterovNODE on the continuous normalizing flow tasks in [Grathwohl et al.(2018)] and times-series regression tasks as you suggested. We will include these new results in the discussion and in the updated version of our manuscript.
> >
> > Regarding your concern of using NesterovNODE for time-series, please let us address it here. The NesterovNODE solves the Nesterov ODE given by Eq. 8 in our paper. This Nesterov ODE is the continuous-time limit of the Nesterov’s scheme in Eq. 7. The second equation in Eq. 7 accumulates the updates $x_{k} - x_{k-1}$, smoothing out the updates by an exponential moving average with a time-dependent smoothing coefficient. The damping term in the Nesterov ODE is derived from this exponential moving average, and the exponential moving average is popularly used in modeling time-series including the autoregressive moving average model (ARMA) and the autoregressive integrated moving average model (ARIMA) [Box et al.(1994)]. Also, damping is considered in many physical systems and accounts for the processes that dissipate the energy stored in the oscillation. Examples include viscous drag in mechanical systems and resistance in electronic oscillators. Thus, having damping included in the ODE is not undesirable in learning a time-series.
> >
> > For the applications such as modeling orbiting planets in which we know there is no damping present, then we can include this domain knowledge in developing the new NODE model that fits the application better. However, it might not be entirely true that there is no damping present in orbiting planets if we consider the gravitational field. The general relativity might suggest that any moving matter has a damping effect in the gravitational field. For example, ​​two planets orbiting each other can emit gravitational waves. Over time, the orbits of planets will decay due to this gravitational radiation [Peters (1964)].
> >
> > **References**
> >
> > [1] Grathwohl, W., Chen, R.T., Bettencourt, J., Sutskever, I. and Duvenaud, D., 2018. Ffjord: Free-form continuous dynamics for scalable reversible generative models. ICLR. 2018
> >
> > [2] Box, George; Jenkins, Gwilym M.; Reinsel, Gregory C. (1994). Time Series Analysis: Forecasting and Control (Third ed.). Prentice-Hall. ISBN 0130607746.
> >
> > [3] Peters, P.C. Gravitational radiation and the motion of two point masses. Physical Review, 136(4B), p.B1224.1964.

---

> > > ### Comment · Reviewer_hkAe · 2022-08-08
> > > **Is there an update on the additional experiments?**
> > >
> > > I was just wondering if there was an update on the additional experiments that have been running? Or if I missed it? I would consider raising my score further if these were to be provided in the next two days.

---

> > > > ### Author Response · Authors · 2022-08-08
> > > > **Update on the additional experiments**
> > > >
> > > > Thanks again for your endorsement. We have included a new experiment on the **Human Activity dataset** [Lustrek et al. (2010)] in Appendix E.6 of our revision. This dataset consists of time series from five individuals doing various activities: walking, sitting, lying, etc. The task is to classify each time point into one of seven types of activities (walking, sitting, etc.). We use the NODE-RNN, i.e. the ODE-RNN model in [Rubanova et al.(2019)], as the baseline model. We observe that our GNesterovNODE-RNN outperforms the baseline NODE-RNN and GHBNODE-RNN in both test accuracy and NFEs. We summarize our results in Table 8 and Figure 18 in Appendix E.6 of the revision.
> > > >
> > > > We are finishing up another experiment to verify the advantage of the NesterovNODE on continuous normalizing flows for MNIST [Grathwohl et al.(2018)]. We will include this new result in the discussion here and in the updated version of our manuscript before the author-reviewer discussion period ends.
> > > >
> > > > **References**
> > > >
> > > > [1] Lustrek, M. et al. Localization data for person activity data set, 2010.
> > > >
> > > > [2] Rubanova, Y. et al. Latent ordinary differential equations for irregularly-sampled time series. NeurIPS, 2019.
> > > >
> > > > [3] Grathwohl, W. et al.. Ffjord: Free-form continuous dynamics for scalable reversible generative models. ICLR, 2019.

---

> > > > > ### Comment · Reviewer_hkAe · 2022-08-08
> > > > > **Thank you for clarifying**
> > > > >
> > > > > Thank you for clarifying, apologies if I missed that. If there is an update on the CNFs, it would be great to say here so I can reevaluate the initial score.
> > > > >
> > > > > In the camera-ready version of this paper, it may be worthwhile running all of the baselines on this Human Activity Task, I appreciate that is hard during a one week rebuttal period.

---

> > > > > > ### Author Response · Authors · 2022-08-09
> > > > > > **Update on the continuous normalizing flows for MNIST**
> > > > > >
> > > > > > Thanks for your further feedback, and we appreciate your endorsement. We will run all the baselines on the Human Activity and CNFs tasks in the camera-ready version of the paper.
> > > > > >
> > > > > > As promised in our previous reply, we have conducted an experiment on the continuous normalizing flows for MNIST and included the results in Figure 19 and Table 9 in Appendix E.7 of our revision. We summarize our results below. The continuous normalizing flow model we use is the FFJORD model in [Grathwohl et al.(2019)], and we perform variational inference using the FFJORD models with NODE, GHBNODE, and GNesterovNODE. The setup of our experiment follows Section 4.3 in [Grathwohl et al.(2019)]. In this task, the GNesterovNODE significantly reduces the NFEs in both forward and backward passes while improving the negative ELBO on the test set compared to the NODE and GHBNODE.
> > > > > >
> > > > > > Table 1: The FFJORD-GNesterovNODE vs. the baseline FFJORD-NODE and FFJORD-GHBNODE for use in variational inference with a continuous normalizing flow model, i.e. FFJORD [Grathwohl et al., 2019a], on the binarized MNIST dataset. We also include the reported results from [Grathwohl et al., 2019a] (in parentheses) in addition to our reproduced results.
> > > > > >
> > > > > > | Method      | Average Forward NFEs  | Average Backward NFEs   | Negative ELBO
> > > > > > | :---        |    :----:   |   :----:   | :----:   |
> > > > > > | FFJORD-NODE    |  97.92   |  100.76  | 88.30 (82.82)  |
> > > > > > | FFJORD-GHBNODE     |  90.33     |  98.77  | 75.87  |
> > > > > > | FFJORD-GNesterovNODE     |    **62.04**   | **51.07**   | **72.16**  |
> > > > > >
> > > > > >
> > > > > > **References**
> > > > > >
> > > > > > [1] Grathwohl, W. et al.. Ffjord: Free-form continuous dynamics for scalable reversible generative models. ICLR, 2019.

---

> > > > > > > ### Comment · Reviewer_hkAe · 2022-08-09
> > > > > > > **Thank you, I raise my score**
> > > > > > >
> > > > > > > Thank you very much for carrying out that experiment. I will raise my score accordingly. I would suggest that for the final version of this paper the CNF and time-series experiments are made more prominent.

---

> > > > > > > > ### Author Response · Authors · 2022-08-09
> > > > > > > > **Thanks for your endorsement!**
> > > > > > > >
> > > > > > > > Thanks for your suggestion and we appreciate your endorsement. We will follow your suggestion and make the CNF and time-series experiments more prominent in the final version of our paper.

---

### Official Review · Reviewer_tT37 · 2022-07-08

**Rating:** 6
**Confidence:** 4
**Soundness:** 2 fair
**Presentation:** 3 good
**Contribution:** 2 fair

**Summary:**

The paper presents the novel class of neural ODEs called NesterovNODEs, where the ODE block solves the second-order ODE corresponding to the limit of the Nesterov accelerated gradient scheme. To avoid instability around $t=0$ in this second-order ODE, the authors propose the generalized version of NesterovNODEs, where the variable transformation eliminates this instability. The main benefit of this modification of the neural ODE approach is a significant reduction of the NFE needed to perform forward and backward passes. At the same time, the test quality of considered models is improved compared to other neural ODE models. This gain is shown point cloud separation, image classification, and sequence modeling problems.

**Questions:**

The following typos should be fixed in the revised version
1) lines 151,153 - bold notation for $m(t)$
2) line 151 - please provide explicit form for activation function $\sigma$.
3) line 314 $1/k^2$

Also, please comment the following questions and remarks
1) It is known that adjoint method to compute the gradients through the solution of ODE is unstable, see e.g.
`Gholami, A., Keutzer, K., & Biros, G. (2019). Anode: Unconditionally accurate memory-efficient gradients for neural odes. arXiv preprint arXiv:1902.10298` How did you control the potential instabilities in the adjoint method for NesterovODE? Authors addressed the vanishing gradient problem, but what about exploding gradient problem?
2) Please add more details about Figure 2: what step size and ODE solver is used for forward and backward passes?
3) The role of $T - t$ in line 170 should be explicitly explained for the reader convenience
4) Please specify the initial value in Eq 15 and 17. Otherwise, it is unclear where the integration is forward in time and where it is backward in time?
5) Please add the main conclusion from these plots in caption of Fig 3
6) Please explicitly define $v$ as a row in Proposition 3
7) Please fix legend in Figure 9
8) Please comment the difference between NesterovNode and GNesterovNode demonstrated in Figure 3. From this plot follows that HBNode is the best in terms of treating vanishing gradient problem.

**Limitations:**

The missing limitations of this work is increasing dimension in three times (store $x(t), h(t)$ and $m(t)$) of the NesterovNODE compared with standard neural ODE architecture (only $x(t)$ is needed). Such increasing can make the proposed approach intractable for large dimension of hidden state, so discussion and probably experiments on the memory consumption should be added in the revised version.

**Strengths And Weaknesses:**

Strengths
1) the presentation of the paper is clear and easy to follow
2) the extensive computational experiments confirm expectations from the presented theoretical part
3) the proposed method has theoretical explanations

Weaknesses
1) the used model in the image classification problem gives too low test accuracy. The papers

a) Zhang, T., Yao, Z., Gholami, A., Gonzalez, J. E., Keutzer, K., Mahoney, M. W., & Biros, G. (2019). ANODEV2: A coupled neural ODE framework. Advances in Neural Information Processing Systems, 32.

b) Gusak, J., Markeeva, L., Daulbaev, T., Katrutsa, A., Cichocki, A., & Oseledets, I. (2020, April). Towards Understanding Normalization in Neural ODEs. In ICLR 2020 Workshop on Integration of Deep Neural Models and Differential Equations.

c) Daulbaev, T., Katrutsa, A., Markeeva, L., Gusak, J., Cichocki, A., & Oseledets, I. (2020). Interpolation technique to speed up gradients
propagation in neural ODEs. Advances in Neural Information Processing Systems, 33, 16689-16700.

show the +80% test accuracy in the CIFAR10 with standard neural ODE model without any second-order modifications.
So the main concern is that the proposed approach is tested only on the continuous models that do not provide state-of-the-art accuracy for the considered problems.

2) the original motivation for NAG is to solve optimization problem faster and authors provide clear illustration of such convergence in Figure 1. At the same time, the motivation of continuous models like neural ODEs is to fit hidden dynamics in the latent space to learn relevant features. From these motivations is not clear why the ODE that is continuous limit of NAG is a good choice for the ODE that should learn hidden dynamics in continuous model.

3) Figure 3 contradicts the claim about absence of vanishing gradient in both NesterovNODE-RNN and GNesterovNODE-RNN since the plots for them demonstrate the different trends.

4) From Table 3 in Appendix follows that for the different considered methods (NODE, ANODE,NODE, etc) different models are tested. If so, we can not split effect of different architectures and effect of different method.

---

> ### Author Response · Authors · 2022-08-02
> **Response to Reviewer tT37 (1)**
>
> Thank you for your thoughtful review and valuable feedback. Below we address your concerns.
>
> -----
>
> **Q1. The original motivation for NAG is to solve optimization problem faster and authors provide clear illustration of such convergence in Figure 1. At the same time, the motivation of continuous models like neural ODEs is to fit hidden dynamics in the latent space to learn relevant features. From these motivations is not clear why the ODE that is continuous limit of NAG is a good choice for the ODE that should learn hidden dynamics in continuous model.**
>
> **Reply:** The continuous limit of NAG is also an ODE. When replacing the gradient term in Eq. 5 in our paper by a neural network, we turn this ODE limit into a second-order NODE, the NesterovNODE in Eq. 6, which like other NODEs, can fit the hidden dynamics in the latent space to learn relevant features by learning the parameters of the neural network $f(\boldsymbol{x}(t), t, \theta)$ in Eq. 6 from the training data. In the meantime, compared to the baseline first-order NODE, the NesterovNODE converges faster, reduces the NFEs, and thus improves the model’s efficiency.
>
> **Q2. Figure 3 contradicts the claim about absence of vanishing gradient in both NesterovNODE-RNN and GNesterovNODE-RNN since the plots for them demonstrate the different trends.Please comment the difference between NesterovNode and GNesterovNode demonstrated in Figure 3. From this plot follows that HBNode is the best in terms of treating vanishing gradient problem.**
>
> **Reply:** We believe there is a misunderstanding of the results reported in Figure 3. Please allow us to clear this misunderstanding by explaining the results from Figure 3.  Figure 3 plots of the L2-norm of the adjoint states for the NODE-RNN, the GHBNODE-RNN, the NesterovNODE-RNN and the  GNesterovNODE-RNN back-propagated from the last time stamp. The adjoint state of NODE-RNN vanishes quickly when the gap between the final time $T$ and intermediate time $t$ becomes larger, while the adjoint states of the NesterovNODE-RNN and the GNesterovNODE-RNN, as well as the GHBNODE-RNN, decay much more slowly. This implies that NesterovNODE-RNN and GNesterovNODE-RNN are more effective in learning long-term dependency than NODE-RNN.
>
> Note that the GNesterovNODE is the NesterovNODE with an additional activation function $\sigma$ applied to the function $f(\boldsymbol{h}(t), t, \theta)$, the factor $t^{-3/2}e^{t/2}$, and the momentum state $\boldsymbol{m}(t)$. In addition, in the GNesterovNODE, an additional residual term $\xi \boldsymbol{h}(t)$ is added. These modification in the GNesterovNODE is to prevent the potential blow-up of $\boldsymbol{h}(t)$.
>
> From Figure 3, GHBNODE avoids the vanishing gradient problem better than NesterovNODE and GNesterovNODE. However, NesterovNODE and GNesterovNODE require much fewer NFEs. i.e. being more efficient, than the GHBNODE while yielding better accuracy than the GHBNODE in our experiments (see Figure 4, 5, 6, 11 and Table 1 in the paper).
>
> **Q4. From Table 3 in Appendix follows that for the different considered methods (NODE, ANODE,NODE, etc) different models are tested. If so, we can not split effect of different architectures and effect of different method.**
>
> **Reply:** The difference in the architecture between first-order NeuralODE methods (NODE, ANODE) and second-order NeuralODE methods (SONODE, HBNODE/GHBNODE, NesterovNODE/GNesterovNODE) happens because of the structural difference in their dynamics function. In particular, second-order NeuralODE methods use extra states to model the momentum of the original hidden states. To get roughly similar numbers of parameters for first-order NeuralODE methods and second-order NeuralODE methods, there must be differences in their architecture. In choosing the architecture for our method, we have made sure that there is no meaningful architecture difference in our methods compared to SONODE and HBNODE/GHBNODE.
>
> **Q5. The used model in the image classification problem gives too low test accuracy. The papers ... show the +80% test accuracy in the CIFAR10 with standard neural ODE model without any second-order modifications. So the main concern is that the proposed approach is tested only on the continuous models that do not provide state-of-the-art accuracy for the considered problems.**
>
> **Reply:** Thank you for your suggestion. We are working on using these papers as baselines to perform more experiments. We will update in the discussion week.
>
> **Q6. line 151 - please provide explicit form for activation function $\sigma$.**
>
> **Reply:** The activation function can be any activation function commonly used, so we do not specify one here. In our experiments, we use $tanh$ and $hardtanh$ for $\sigma$.

---

> > ### Author Response · Authors · 2022-08-02
> > **Response to Reviewer tT37 (2)**
> >
> > **Q7. It is known that adjoint method to compute the gradients through the solution of ODE is unstable, see e.g. Gholami, A., Keutzer, K., & Biros, G. (2019). Anode: Unconditionally accurate memory-efficient gradients for neural odes. arXiv preprint arXiv:1902.10298. How did you control the potential instabilities in the adjoint method for NesterovODE? Authors addressed the vanishing gradient problem, but what about exploding gradient problem?**
> >
> > **Reply:** The exploding gradients issue can be effectively resolved via gradient clipping, training loss regularization, etc [1, 2]. Thus in practice the vanishing gradient is the major issue for learning long-term dependencies [1]. We have addressed this in Remark 3 in Section 4 of our revision.
> >
> > [1] Razvan Pascanu, Tomas Mikolov, and Yoshua Bengio. On the difficulty of training recurrent neural networks. In International Conference on Machine Learning, pages 1310–1318, 2013.
> >
> > [2] N. Benjamin Erichson, Omri Azencot, Alejandro Queiruga, Liam Hodgkinson, and Michael W. Mahoney. Lipschitz recurrent neural networks. In International Conference on Learning Representations, 2021.
> >
> > **Q8.  Please add more details about Figure 2: what step size and ODE solver is used for forward and backward passes?**
> >
> > **Reply:** In this experiment, we use the $\texttt{dopri5}$ solver (implemented in $\texttt{torchdiffeq}$) with a tolerance of $10^{-7}$ for adaptive step sizes in both forward and backward passes. We have addressed this at the start of Appendix D of our revision.
> >
> > **Q9. The role of T−t  in line 170 should be explicitly explained for the reader convenience**
> >
> > **Reply:**
> > The term $T-t$ demonstrates the gap between the final time $T$ and intermediate times $t$. As can be seen from Fig. 3, when this term becomes larger, the adjoint states of NesterovNODE-RNN, GNesterovNODE-RNN, and GHBNODE-RNN decay much more slowly than NODE-RNN. Thus, NesterovNODE-RNN and GNesterovNODE-RNN have the ability to tackle the vanishing gradient issue. We have clarified the role of $T-t$ in the revision. We have addressed this in the caption of Figure 3 in our revision.
> >
> > **Q10. Please specify the initial value in Eq 15 and 17. Otherwise, it is unclear where the integration is forward in time and where it is backward in time?**
> >
> > **Reply:** We have specified the initial value conditions of Eqs. (8) and (15) and the final value conditions of Eq. (17) in the revised version. It is worth noting that, in practice, we use other neural networks to learn more appropriate initial values from the raw inputs. These neural networks can be dense networks or CNNs depending on the input type. After that, we pass the learned initial values to the ODE layer.
> >
> > **Q11. Typos: The following typos should be fixed in the revised version... Please add the main conclusion from these plots in caption of Fig 3. Please explicitly define $v$ as a row in Proposition 3. Please fix legend in Figure 9.**
> >
> > **Reply:** Thank you for your suggestion. We have updated these details in our revision.
> >
> >
> > **Q12. The missing limitations of this work is increasing dimension in three times (store x(t),h(t) and m(t)) of the NesterovNODE compared with standard neural ODE architecture (only x(t) is needed). Such increasing can make the proposed approach intractable for large dimension of hidden state, so discussion and probably experiments on the memory consumption should be added in the revised version.**
> >
> > **Reply:** One of the key advantages of NODEs is that when the continuous adjoint method is used to compute the gradients in the backward step, the memory complexity of the model is $O(1)$, i.e., constant memory cost with respect to the number of layer [Chen et al.(2018)] since we do not need to store intermediate quantities of the forward pass. NesterovNODE takes advantage of this constant memory cost to introduce two additional state variables, $\boldsymbol{h}(t)$ and $\boldsymbol{m}(t)$.
> > We include our memory consumption analysis below and in Appendix E.3 of the revision. Memory consumption numbers are extracted using the function max_memory_allocated in PyTorch on the CIFAR10 dataset (https://pytorch.org/docs/stable/generated/torch.cuda.max_memory_allocated.html).
> >
> > | Method      | Maximum Memory Consumption (GB) |
> > | :---        |    :----:   |
> > | NODE    |  3.2046  |
> > | ANODE     |    2.3292   |
> > | SONODE     |    2.2034   |
> > | HBNODE     |   2.2346   |
> > | GHBNODE     |   2.2456   |
> > | NesterovNODE     |  2.2346   |
> > | GNesterovNODE     |  ​​2.2580  |
> >
> > **References**
> >
> > [1] Ricky TQ Chen, Yulia Rubanova, Jesse Bettencourt, and David Duvenaud. Neural ordinary differential equations. In Proceedings of the 32nd International Conference on Neural Information Processing Systems, pages 6572–6583, 2018.
> >
> > ---
> > We hope we have cleared your concerns about our work. We have also revised our manuscript according to your comments, and we would appreciate it if we can get your further feedback at your earliest convenience.

---

> > > ### Comment · Area_Chair_mi6M · 2022-08-07
> > > **Any feedback?**
> > >
> > > Dear Reviewer tT37, the authors have provided a feedback. Any chance for a "live" discussion? I find it very interesting in productive in most cases (rather that question-answer thing).

---

> > > > ### Author Response · Authors · 2022-08-07
> > > > **Thanks for Fostering Discussion!**
> > > >
> > > > We would like to thank the ACs for leading and promoting the discussion of our paper. We really appreciate it. We are finishing up additional experiments to verify the advantage of the NesterovNODE on continuous normalizing flows and times-series regression tasks as the reviewers suggested. We will include these new results in the discussion here and in the updated version of our manuscript before the author-reviewer discussion period ends.

---

> > > ### Author Response · Authors · 2022-08-09
> > > **Response to Reviewer tT37 - Additional Experiments for Time-series and Continuous Normalizing Flows**
> > >
> > > Dear reviewer,
> > >
> > > In order to address your concern that  "the proposed approach is tested only on the continuous models that do not provide state-of-the-art accuracy for the considered problems," **we have run additional experiments for the Human Activity time-series classification [Lustrek et al. (2010)] and the continuous normalizing flow task for MNIST with strong baseline models** to verify the advantage of our GNesterovNODE. On Human Activity time-series classification, our baseline model is the ODE-RNN in [Rubanova et al.(2019)], and on the continuous normalizing flow task for MNIST, our baseline model is the FFJORD in [Grathwohl et al.(2019)]. GNesterovNODE helps improve the accuracy and reduce the NFEs in these benchmark experiments. We summarize our results below. We have also included the results for Human Activity time-series classification in in Table 8 and Figure 18 in Appendix E.6 of our revision. Results for the continuous normalizing flows for MNIST are provided in Table 9 and Figure 19 in Appendix E.7 of our revision.
> > >
> > > We would appreciate it if you could let us know if there are additional questions or concerns about our revision and rebuttal. We would be happy to do any follow-up discussion or address any additional comments.
> > >
> > > Table 1: The FFJORD-GNesterovNODE vs. the baseline FFJORD-NODE and FFJORD-GHBNODE for use in variational inference with a continuous normalizing flow model, i.e. FFJORD [Grathwohl et al., 2019a], on the binarized MNIST dataset. We also include the reported results from [Grathwohl et al., 2019a] (in parentheses) in addition to our reproduced results.
> > >
> > > | Method      | Average Forward NFEs  | Average Backward NFEs   | Negative ELBO
> > > | :---        |    :----:   |   :----:   | :----:   |
> > > | FFJORD-NODE    |  97.92   |  100.76  | 88.30 (82.82)  |
> > > | FFJORD-GHBNODE     |  90.33     |  98.77  | 75.87  |
> > > | FFJORD-GNesterovNODE     |    **62.04**   | **51.07**   | **72.16**  |
> > >
> > >
> > > Table 2: The GNesterovNODE-RNN vs. the baseline NODE-RNN and GHBNODE-RNN on the Human Activity benchmark [Lustrek et al. (2010)].
> > >
> > > | Method      | Average Forward NFEs  | Average Backward NFEs   | Test Accuracy (%)
> > > | :---        |    :----:   |   :----:   | :----:   |
> > > | NODE-RNN    |  220.74   |  396.42  | 82.90  |
> > > | GHBNODE-RNN     |  51.85     |  221.26  | 83.80  |
> > > | GNesterovNODE-RNN     |    **46.44**   | **210.84**   | **84.00**  |
> > >
> > >
> > > **References**
> > >
> > > [1] Lustrek, M. et al. Localization data for person activity data set, 2010.
> > >
> > > [2] Rubanova, Y. et al. Latent ordinary differential equations for irregularly-sampled time series. NeurIPS, 2019.
> > >
> > > [3] Grathwohl, W. et al.. Ffjord: Free-form continuous dynamics for scalable reversible generative models. ICLR 2019.

---

> > > > ### Comment · Reviewer_tT37 · 2022-08-09
> > > > **Feedback**
> > > >
> > > > Dear authors,
> > > >
> > > > thank you for additional experiments and empirical evaluations of the memory consumption! They significantly improve the quality of the submission and make the proposed approach more convincing and competitive with state-of-the-art approaches to solve the relevant problems. I have increased the score.

---

> > > > > ### Author Response · Authors · 2022-08-09
> > > > > **Thanks for your endorsement!**
> > > > >
> > > > > Thanks for your response and we appreciate your endorsement.

---

### Official Review · Reviewer_Uc87 · 2022-07-09

**Rating:** 7
**Confidence:** 4
**Soundness:** 3 good
**Presentation:** 3 good
**Contribution:** 2 fair

**Summary:**

NesterovNODE and GNesterovNODE are introduced, as an alternate DAE-based parameterisation of a neural ODE. They are explored primarily for their ability to improve accuracy and reduce computational cost.

**Questions:**

While the bulk of my review touches on several points, I would particularly highlight the following questions/suggestions:

1.  I quite like the trick of reparameterising from $h$ to $x$ in order to obtain a singularity at $t=0$. I am curious what motivated this approach in particular; how was this derived? And as above, I am curious why this approach is taken, instead of simply starting the integration from e.g. $t=1$ rather than $t=0$.
2. As above, how is the numerics/software was handled?
3. The experimental weaknesses offer substantial room for improvement.

**Limitations:**

There are no ethical concerns with this paper.

**Strengths And Weaknesses:**

**Strength: topic**

The topic at hand is a meaningful one, as good parameterisations of neural ODEs is still an underexplored problem.

**Strength: Nesterov/DAE approach**

I like the approach of using Nesterov's method as a starting point for choosing a good architecture. This seems well-motivated, and the constructions presented both plausible and novel. (To the best of my knowledge.)

In addition, I like the formulation of the problem as a DAE. To the best of my knowledge "neural DAEs" have received essentially no attention whatsoever, so this is a very interesting line of work to pursue.

**Strength: presentation**

The presentation is at all times both clear and technically precise. I greatly appreciated this aspect of the paper, which I enjoyed reading.

**Strength: experiments**

Several different kinds of experiments are done, and in each case a careful analysis is performed. These are generally well done.

**Weakness: handling blow-up**

The authors go to quite some effort mitigating the blow-up of the Nesterov equations. I feel like much simpler approaches are available -- for example, starting the integration from $t=1$ rather than $t=0$.

**Weakness: comparison to gradient descent**

Lines 81--88: I strongly object to the claim that neural ODEs "implicitly perform gradient descent". It is trivially true that both NODEs and GD are cases of generic ODEs; that is all that is going on here.

**Weakness: "adjoint sensitivity"**

The term "adjoint" has become heavily overloaded. There are multiple ways to backpropagate through a differential equation solve. In the current literature is is often used to mean specifically optimise-then-discretise, but in older literature it was often used to mean specifically discretise-then-optimise. Following [0, Remark 5.5] I recommend avoiding the term "adjoint" altogether.

In passing, since in this work backpropagation is performed via OtD, then the authors may find speedups by utilising the technique of [1].

**Weakness: numerical/software**

It is claimed that Dormand--Prince 5(4) was used to solve the DAE problems presented. However DP5(4) is an ODE solver, not a DAE solver. The problem can of course be rewritten in ODE form (e.g. equation 10) but this would no longer seem to follow the DAE presentation that is otherwise adhered to. How were the numerical methods actually implemented, i.e. what ODE was actually solved?

I note that for example [2] implements DAE solvers directly, which could have been an alternate approach.

In passing, note that DP5(4) is now generally reckoned as being slightly less efficient than the Tsitouras 5(4) method [3], which can be found implemented in both [2] and [4].

I do not see the choice of software listed anywhere. PyTorch with torchdiffeq, JAX with Diffrax, Julia with DifferentialEquations.jl, or something else?

Finally, I do not see any discussion on the choice of tolerance, step size controller, etc. (Was this DP5(4) with a fixed step size? If so, what step size? Was it with an adaptive step size controller? If so, which controller and what tolerances?)

**Weakness: experiments**

I would suggest removing the MNIST experiments. MNIST is linearly separable at (as I recall) 93% accuracy. It is not a meaningful benchmark.

Moreover, I do not think image classification is much of an interesting benchmark in the first place. I do not think anyone would claim that NDEs are the appropriate tool for performing image classification: standard ResNets/Transformers/etc. are much easier to use on this problem, for which the continuous-time formulation confers little advantage.

Following [0, Section 1.2.1 and Section 2.2.1] I would strongly suggest targeting other applications (time series, scientific problems, etc.) instead.

The time series experiments seem very strange to me. There is no discussion on how the time series is provided as an input to the ODE. There are numerous approaches here, mostly based around either:

1. As a controlled differential equation [5]
2. Discrete updates, e.g. an ODE-RNN [Ref. Rubanova et al. 2019 from this paper]

In either case the ODE formulation must be changed quite substantially. If working in continuous time (e.g. as in [5]) then there is an additional question of how the discrete data is converted (by interpolation or otherwise) into continuous time [6]. Consulting the appendix it looks like specifically an ODE-RNN approach is used? As there has been substantial work on this topic then much more detail deserves to be in the main paper.

**Weakness: various minor issues**

Figure 1: This relationship between GD, HB and NAG is well-known, and does not need to be here.

Remark 2: it is not necessary that specifically the Euler method be used. This remark should be reworded: "If the Euler method is used, then ... [the rest of the remark follows]".

GNesterovNODE: I'm not sure the term "generalised" is the appropriate one here -- this seems to have been modified, not generalised.

Long-term dependencies: I am used to think of "long term dependencies" as being in referenced to long time series, e.g. in the context of RNNs or neural CDEs. As the present paper concerns itself with non-time-series problems, I would suggesting focusing on the terminology of "vanishing gradients" etc. instead. (In terms of technical content, I did like this section of the paper.)

Rubanova et al. 2019 (the Latent ODE paper) appears in the reference list twice.

**Commentary**

I suspect there are natural extensions to this paper, studying neural DAEs more generally.

The fact that increasing $r$ increases both NFEs and model accuracy suggests to me that this may be due to discretisation error: greater NFEs produces a larger overall computation graph and thus increased model capacity. If the tolerances are shrunk small enough then I speculate that this improvement should vanish, as the model converges to the true solution of the ODE.

**References**

[0] On Neural Differential Equations, Kidger, Doctoral Thesis, University of Oxford 2021

[1] "Hey, that's not an ODE": Faster ODE Adjoints via Seminorms, Kidger et al, ICML 2021

[2] DifferentialEquations.jl, https://diffeq.sciml.ai/

[3] Runge--Kutta pairs of order 5 (4) satisfying only the first column simplifying assumption, Tsitouras, Computers and Mathematics with Applications 2011

[4] Diffrax, https://github.com/patrick-kidger/diffrax

[5] Neural Controlled Differential Equations for Irregular Time Series, Kidger et al., NeurIPS 2020

[6] Neural Controlled Differential Equations for Online Prediction Tasks, Morrill et al., arXiv:2106.11028

---

> ### Author Response · Authors · 2022-08-02
> **Response to Reviewer Uc87 (1)**
>
> Thank you for your thoughtful review and valuable feedback. Below we address your concerns.
>
> -----
>
> **Q1. Handling blow-up:  The authors go to quite some effort mitigating the blow-up of the Nesterov equations. I feel like much simpler approaches are available -- for example, starting the integration from t=1 rather than t=0. I quite like the trick of reparameterising from h to x in order to obtain a singularity at t=0. I am curious what motivated this approach in particular; how was this derived? And as above, I am curious why this approach is taken, instead of simply starting the integration from e.g. t=1 rather than t=0.**
>
> **Reply:** We reparametrize from h to x in order to eliminate the time-dependent damping coefficient $3/t$ in the Nesterov ODE given by Eq. 8 in our paper. In particular, we set $h(t) = k(t)x(t)$ and then find $k(t)$ such that Eq. 8 written in terms of $x(t)$ matches the Heavy-Ball ODE with a constant damping coefficient given by Eq. 5 in our paper. We find that $k(t) = t^{-3/2}e^{t/2}$ satisfies this requirement. Solving this Heavy-Ball ODE with respect to $x(t)$ is easier and more stable than solving the Nesterov ODE with respect to $h(t)$.
>
> We have tried to start the integration from $t = 1$ as the reviewer suggests. However, our empirical results on CIFAR10 and Walker2d benchmark show that this approach shows much worse accuracy and efficiency than using our reparametrization trick. We include those results in Appendix E.1 of our revision.
>
> **Q2. Comparison to gradient descent Lines 81--88: I strongly object to the claim that neural ODEs "implicitly perform gradient descent". It is trivially true that both NODEs and GD are cases of generic ODEs; that is all that is going on here.**
>
> **Reply:** We agree with the reviewer and have changed the title of that paragraph to “ODE limit of gradient descent and connection to NODEs” in our revision.
>
> **Q3.  Adjoint sensitivity The term "adjoint" has become heavily overloaded. There are multiple ways to backpropagate through a differential equation solver. In the current literature it is often used to mean specifically optimise-then-discretise, but in older literature it was often used to mean specifically discretise-then-optimise. Following [0, Remark 5.5] I recommend avoiding the term "adjoint" altogether.
> In passing, since in this work backpropagation is performed via OtD, then the authors may find speedups by utilising the technique of [1].**
>
> **[0] On Neural Differential Equations, Kidger, Doctoral Thesis, University of Oxford 2021.**
>
> **[1] "Hey, that's not an ODE": Faster ODE Adjoints via Seminorms, Kidger et al, ICML 2021.**
>
> **Reply:** Thanks for your clarification and suggestions. In this paper, we use the optimise-then-discretise approach. Following [0], we refer to it as the “continuous adjoint method” in our revision. We have also added a discussion about [1] and how to use their proposed techniques to speed up our NesterovNODEs in the Related Work section. We have also applied the seminorm method in [1] to the GNesterovNODEs trained for CIFAR10. We include this result in Appendix E.5 of our revision.

---

> > ### Author Response · Authors · 2022-08-02
> > **Response to Reviewer Uc87 (2)**
> >
> > **Q4. Numerical/software: It is claimed that Dormand--Prince 5(4) was used to solve the DAE problems presented. However DP5(4) is an ODE solver, not a DAE solver. The problem can of course be rewritten in ODE form (e.g. equation 10) but this would no longer seem to follow the DAE presentation that is otherwise adhered to. How were the numerical methods actually implemented, i.e. what ODE was actually solved?
> > I note that for example [2] implements DAE solvers directly, which could have been an alternate approach.
> > In passing, note that DP5(4) is now generally reckoned as being slightly less efficient than the Tsitouras 5(4) method [3], which can be found implemented in both [2] and [4].
> > As above, how is the numerics/software was handled?**
> >
> > **[2] DifferentialEquations.jl, https://diffeq.sciml.ai/**
> >
> > **[3] Runge--Kutta pairs of order 5 (4) satisfying only the first column simplifying assumption, Tsitouras, Computers and Mathematics with Applications 2011**
> >
> > **[4] Diffrax, https://github.com/patrick-kidger/diffrax**
> >
> > **Reply:** Thank you for your suggestions. To solve the NesterovNODE in Eq. 9, we rewrite it in the differential-algebraic form given by Eq. 14. We solve Eq. 14 as follows:
> >
> > Step 1: Given $h(t_0)$, we compute $x(t_0)$.
> >
> > Step 2: Using the computed $x(t_0)$ in Step 1, we solve the the ODE given by the last two equations in 14, $x'(t) = m(t)$ and $m'(t) = -m(t) - f(h(t), t, \theta)$ by using the Dormand--Prince 5(4) solver. To get $h(t)$ for the second differential equation, we compute $h(t)$ from $x(t)$. Alternatively, the second differential equation can be expressed as $m'(t) = -m(t) - f(x(t), t, \theta)$, which we can think of applying a composite function $f(g(h(t), t, \theta)$ to $h(t)$.
> >
> > Step 3: Compute $h(t_n)$ from $x(t_n)$.
> >
> > We have also added a discussion on DAE solvers in [2] and Tsitouras 5(4) method in [2, 3, 4] in Appendix F of our revision.
> >
> > **Q5. Numerical/software: I do not see the choice of software listed anywhere. PyTorch with torchdiffeq, JAX with Diffrax, Julia with DifferentialEquations.jl, or something else?
> > Finally, I do not see any discussion on the choice of tolerance, step size controller, etc. (Was this DP5(4) with a fixed step size? If so, what step size? Was it with an adaptive step size controller? If so, which controller and what tolerances?)**
> >
> > **Reply:** We implemented the models using PyTorch with $\texttt{torchdiffeq}$. The solver we used is the default $\texttt{dopri5}$ with adaptive step sizes in $\texttt{torchdiffeq}$ (except for the experiment in Section 6, Remark 2, in which we also use euler, rk4, and explicit adams). We have revised our paper to address these. We have specified the tolerances and the step sizes for the experiments in the corresponding figures’ captions and summarized those in Appendix D of our revision.
> >
> > **Q6. Experiments: I would suggest removing the MNIST experiments. MNIST is linearly separable at (as I recall) 93% accuracy. It is not a meaningful benchmark.
> > Moreover, I do not think image classification is much of an interesting benchmark in the first place. I do not think anyone would claim that NDEs are the appropriate tool for performing image classification: standard ResNets/Transformers/etc. are much easier to use on this problem, for which the continuous-time formulation confers little advantage.
> > Following [0, Section 1.2.1 and Section 2.2.1] I would strongly suggest targeting other applications (time series, scientific problems, etc.) instead.**
> >
> > **Reply:** Thanks for your suggestions. Even though neural differential equations (NDEs) might not be the appropriate tool for performance image classification, image benchmarks such as CIFAR10 and MNIST are still good testbeds to study the behavior and efficiency of a new NDE model. We are currently applying NesterovNODE to other applications as the reviewer suggests and will report those results in the discussion stage.

---

> > > ### Author Response · Authors · 2022-08-02
> > > **Response to Reviewer Uc87 (3)**
> > >
> > > **Q7. Experiments: The time series experiments seem very strange to me. There is no discussion on how the time series is provided as an input to the ODE. There are numerous approaches here, mostly based around either:
> > > As a controlled differential equation [5]
> > > Discrete updates, e.g. an ODE-RNN [Ref. Rubanova et al. 2019 from this paper]
> > > In either case the ODE formulation must be changed quite substantially. If working in continuous time (e.g. as in [5]) then there is an additional question of how the discrete data is converted (by interpolation or otherwise) into continuous time [6]. Consulting the appendix it looks like specifically an ODE-RNN approach is used? As there has been substantial work on this topic then much more detail deserves to be in the main paper.**
> > >
> > > **Reply:** In Section 5.3 in our paper, we use the ODE-RNN framework [7], with the recognition model being set to different ODE-based models, to study Walker2D kinematic simulation task, which requires learning long-term dependency effectively [8]. The dataset [9] consists of a dynamical system from kinematic simulation of a person walking from a pre-trained policy, aiming to learn the kinematic simulation of the MuJoCo physics engine [10]. We randomly take out 10% of the data to make the time series irregularly-sampled. Each input sequence consists of 64 timestamps, which are recurrently fed through a hybrid technique, with the output of the hybrid method being transferred to a single dense layer to form the output time series. The goal is to generate an auto-regressive forecast with an output time series that is as close as the input sequence when shifted one time stamp to the right.
> > >
> > > [7] Yulia Rubanova, Ricky T. Q. Chen, and David K Duvenaud. Latent ordinary differential equations for irregularly-sampled time series. In H. Wallach, H. Larochelle, A. Beygelzimer, F. d'Alché-Buc, E. Fox, and R. Garnett, editors, Advances in Neural Information Processing Systems, volume 32. Curran Associates, Inc., 2019.
> > >
> > > [8] Mathias Lechner and Ramin Hasani. Learning long-term dependencies in irregularly-sampled time series. arXiv preprint arXiv:2006.04418, 2020.
> > >
> > > [9] Greg Brockman, Vicki Cheung, Ludwig Pettersson, Jonas Schneider, John Schulman, Jie Tang, and Wojciech Zaremba. OpenAI Gym, 2016. URL http://arxiv.org/abs/1606.01540. cite arXiv:1606.01540.
> > >
> > > [10] Emanuel Todorov, Tom Erez, and Yuval Tassa. Mujoco: A physics engine for model-based control. In 2012 IEEE/RSJ International Conference on Intelligent Robots and Systems, pages 5026–5033, 2012. doi: 10.1109/IROS.2012.6386109.
> > >
> > >
> > > **Q8. Various minor issues
> > > (1) Figure 1: This relationship between GD, HB and NAG is well-known, and does not need to be here. (2) Remark 2: it is not necessary that specifically the Euler method be used. This remark should be reworded: "If the Euler method is used, then ... [the rest of the remark follows]". (3) GNesterovNODE: I'm not sure the term "generalised" is the appropriate one here -- this seems to have been modified, not generalised. (4) Long-term dependencies: I am used to think of "long term dependencies" as being in referenced to long time series, e.g. in the context of RNNs or neural CDEs. As the present paper concerns itself with non-time-series problems, I would suggesting focusing on the terminology of "vanishing gradients" etc. instead. (In terms of technical content, I did like this section of the paper.). (5) Rubanova et al. 2019 (the Latent ODE paper) appears in the reference list twice.**
> > >
> > > **Reply:** Thanks for pointing these out. We have addressed $(2)$ in our revision.
> > >
> > > *On Point (1)*, according to our presentation of the work, we observe that the differences between GD, HB, and NAG, as well as their behaviors, might not be known by many people. We still keep Figure 1 in our paper so that we can make the paper easier to follow for those people who are not familiar with this result.
> > >
> > > *On Point (3)*, we agree with the reviewer that “modified” can be a better word. For our current revision, in order to avoid confusion during the rebuttal and discussion period, we still use “generalized” in the sense that the GNesterovNODE will become the NesterovNODE when in Eq. 15 in our paper, $\phi(\cdot)$ is an identity function and the coefficient $\xi$ of the skip connections is $0$.
> > >
> > > *On Point (4)*, we agree with the reviewer that focusing on the terminology of "vanishing gradients" is a better option. We have revised our paper to address this.
> > >
> > > *On Point (5)*, we have fixed the reference.

---

> > > > ### Author Response · Authors · 2022-08-03
> > > > **Response to Reviewer Uc87 (4)**
> > > >
> > > > **Q9. Commentary: I suspect there are natural extensions to this paper, studying neural DAEs more generally.**
> > > >
> > > > **Reply:** Thank you for your suggestion. Studying Neural DAEs is indeed an interesting research direction, and we will explore it in future work.
> > > >
> > > >
> > > > **Q10. Commentary: The fact that increasing r increases both NFEs and model accuracy suggests to me that this may be due to discretisation error: greater NFEs produces a larger overall computation graph and thus increased model capacity. If the tolerances are shrunk small enough then I speculate that this improvement should vanish, as the model converges to the true solution of the ODE.**
> > > >
> > > > **Reply:** Thank you for your suggestion. We have included the experiment to verify this in Appendix E.2 of our revision.
> > > >
> > > > ---
> > > > We hope we have cleared your concerns about our work. We have also revised our manuscript according to your comments, and we would appreciate it if we can get your further feedback at your earliest convenience.

---

> > > > > ### Comment · Area_Chair_mi6M · 2022-08-07
> > > > > **Any feedback?**
> > > > >
> > > > > Dear Reviewer Uc87, the authors have provided a feedback. Any chance for a "live" discussion? I find it very interesting in productive in most cases (rather that question-answer thing).

---

> > > > > > ### Author Response · Authors · 2022-08-07
> > > > > > **Thanks for Fostering Discussion!**
> > > > > >
> > > > > > We would like to thank the ACs for leading and promoting the discussion of our paper. We really appreciate it. We are finishing up additional experiments to verify the advantage of the NesterovNODE on continuous normalizing flows and times-series regression tasks as the reviewers suggested. We will include these new results in the discussion here and in the updated version of our manuscript before the author-reviewer discussion period ends.

---

> > > > > ### Comment · Reviewer_Uc87 · 2022-08-07
> > > > > **Response**
> > > > >
> > > > > Thank you to the authors for their detailed rebuttal. I am happy to say that the points I considered significant have been addressed, and as such I have updated my score. (5 -> 7)
> > > > >
> > > > > I would note that I am still not completely satisfied with the discussion on my point "comparison to gradient descent" -- I would insist that the relation between specifically *neural* ODEs and gradinet descent is a very thin one indeed.
> > > > >
> > > > > (And for the authors' own information -- the adaptive stepsize controller implemented in torchdiffeq is an I-controller.)

---

> > > > > > ### Author Response · Authors · 2022-08-09
> > > > > > **Thanks for your endorsement!**
> > > > > >
> > > > > > Thanks for your further feedback and the information on the I-controller in torchdiffeq. We appreciate your endorsement.
> > > > > >
> > > > > > We will incorporate your suggestions into the next version of our manuscript.

---

> > > > > ### Author Response · Authors · 2022-08-09
> > > > > **Update on Additional Experiments for Time-series and Continuous Normalizing Flows**
> > > > >
> > > > > Dear reviewer,
> > > > >
> > > > > As promised, **we have conducted an experiment on the continuous normalizing flows for MNIST** and included the results in Figure 19 and Table 9 in Appendix E.7 of our revision. We summarize our results below. The continuous normalizing flow model we use is the FFJORD model in [Grathwohl et al.(2019)], and we perform variational inference using the FFJORD models with NODE, GHBNODE, and GNesterovNODE. The setup of our experiment follows Section 4.3 in [Grathwohl et al.(2019)]. In this task, the GNesterovNODE significantly reduces the NFEs in both forward and backward passes while improving the negative ELBO on the test set compared to the NODE and GHBNODE.
> > > > >
> > > > > Furthermore, **we have included another new experiment on the Human Activity dataset [Lustrek et al. (2010)]** in Appendix E.6 of our revision. This dataset consists of time series from five individuals doing various activities: walking, sitting, lying, etc. The task is to classify each time point into one of seven types of activities (walking, sitting, etc.). We use the NODE-RNN, i.e. the ODE-RNN model in [Rubanova et al.(2019)], as the baseline model. We observe that our GNesterovNODE-RNN outperforms the baseline NODE-RNN and GHBNODE-RNN in both test accuracy and NFEs. We summarize our results below and in Figure 18 and Table 8 in Appendix E.6 of our revision.
> > > > >
> > > > > Table 1: The FFJORD-GNesterovNODE vs. the baseline FFJORD-NODE and FFJORD-GHBNODE for use in variational inference with a continuous normalizing flow model, i.e. FFJORD [Grathwohl et al., 2019a], on the binarized MNIST dataset. We also include the reported results from [Grathwohl et al., 2019a] (in parentheses) in addition to our reproduced results.
> > > > >
> > > > > | Method      | Average Forward NFEs  | Average Backward NFEs   | Negative ELBO
> > > > > | :---        |    :----:   |   :----:   | :----:   |
> > > > > | FFJORD-NODE    |  97.92   |  100.76  | 88.30 (82.82)  |
> > > > > | FFJORD-GHBNODE     |  90.33     |  98.77  | 75.87  |
> > > > > | FFJORD-GNesterovNODE     |    **62.04**   | **51.07**   | **72.16**  |
> > > > >
> > > > >
> > > > > Table 2: The GNesterovNODE-RNN vs. the baseline NODE-RNN and GHBNODE-RNN on the Human Activity benchmark [Lustrek et al. (2010)].
> > > > >
> > > > > | Method      | Average Forward NFEs  | Average Backward NFEs   | Test Accuracy (%)
> > > > > | :---        |    :----:   |   :----:   | :----:   |
> > > > > | NODE-RNN    |  220.74   |  396.42  | 82.90  |
> > > > > | GHBNODE-RNN     |  51.85     |  221.26  | 83.80  |
> > > > > | GNesterovNODE-RNN     |    **46.44**   | **210.84**   | **84.00**  |
> > > > >
> > > > >
> > > > > **References**
> > > > >
> > > > > [1] Lustrek, M. et al. Localization data for person activity data set, 2010.
> > > > >
> > > > > [2] Rubanova, Y. et al. Latent ordinary differential equations for irregularly-sampled time series. NeurIPS, 2019.
> > > > >
> > > > > [3] Grathwohl, W. et al.. Ffjord: Free-form continuous dynamics for scalable reversible generative models. ICLR 2019.

---

### Author Response · Authors · 2022-08-03
**General Response**

Dear AC and reviewers,

Thanks for your thoughtful reviews and valuable comments, which have helped us improve the paper significantly. We are encouraged by the endorsements that: 1) Using Nesterov's method as a starting point for choosing a good architecture is well-motivated (Reviewer Uc87); 2) Our NesterovNODE is novel (Reviewer Uc87) and has theoretical explanations (Reviewer tT37) with strong theoretical findings (Reviewer hkAe); 3) Our extensive computational experiments confirm expectations from the presented theoretical part (Reviewer tT37, hkAe). We have updated our submission based on the reviewers' feedback, and we have highlighted our revision in blue.

One of the common comments is that the experiments are conducted on problems that Neural ODEs aren't typically used for (image classification and point clouds). We first address this comment here. Our NesterovNODE has two main advantages over other NeuralODE models:

1. NesterovNODE helps reduce the number of NFEs needed for solving the ODEs in both the forward and backward passes, thanks to the quadratic convergence rate of the Nesterov’s scheme.

2. NesterovNODE helps alleviate the vanishing gradient issue during training, thanks to its well-structured spectrum.

The experiments conducted in our paper focus on justifying these two advantages of NesterovNODE. Even though NeuralODEs are not typically used for image classification and point cloud tasks, these tasks can still serve as testbeds to study the behavior and efficiency of a new NeuralODE model. We are currently applying NesterovNODE to other applications including the  continuous normalizing flow tasks as the reviewers suggest and will report those results in the discussion stage.

-----

We are glad to answer any further questions you have on our submission.

---

### Author Response · Authors · 2022-08-03
**Summary of Revision**

Incorporating the comments and suggestions from all reviewers, besides fixing typos and notations, we have made the following main changes in the revised paper.

1. We have added experiments on the Human Activity time-series classification task and the continuous normalizing flow task for MNIST in Appendix E.6 and E.7 of our revision, respectively.

2. We have tried to start the integration from t=1 as the reviewer suggests and included those results in Appendix E.1 of our revision.

3. We have applied the seminorm method in [1] to the GNesterovNODEs trained for CIFAR10. We include this result in Appendix E.5 of our revision.

4. We have added a discussion on DAE solvers in [2] and Tsitouras 5(4) method in [2, 3, 4] in Appendix F of our revision.

5. We have specified the tolerances and the step sizes for the experiments in the corresponding figures’ captions and summarized those in Appendix D of our revision.

6. We have included the experiment to verify the effect of increasing the Nesterov factor $r$ when using small tolerance in Appendix E.2 of our revision.

7. We have discussed the exploding gradient issue in Remark 3 in Section 4 of our revision.

8. We have explained the role of T - t in the caption of Figure 3 in our revision.

9. We have included a memory consumption analysis for the NesterovNODE vs. the other NODE models in Appendix E.3 of the revision.

9. We have provided the wall-clock time comparison between (G)NesterovNODEs and baseline models in Appendix E.4 of our revision.

**References**

[1] "Hey, that's not an ODE": Faster ODE Adjoints via Seminorms, Kidger et al, ICML 2021.

[2] DifferentialEquations.jl, https://diffeq.sciml.ai/.

[3] Runge--Kutta pairs of order 5 (4) satisfying only the first column simplifying assumption, Tsitouras, Computers and Mathematics with Applications 2011.

[4] Diffrax, https://github.com/patrick-kidger/diffrax.

---

### Author Response · Authors · 2022-08-09
**Update on Additional Experiments for Time-series and Continuous Normalizing Flows**

Dear ACs and reviewers,

As promised, **we have conducted an experiment on the continuous normalizing flows for MNIST** and included the results in Figure 19 and Table 9 in Appendix E.7 of our revision. We summarize our results below. The continuous normalizing flow model we use is the FFJORD model in [Grathwohl et al.(2019)], and we perform variational inference using the FFJORD models with NODE, GHBNODE, and GNesterovNODE. The setup of our experiment follows Section 4.3 in [Grathwohl et al.(2019)]. In this task, the GNesterovNODE significantly reduces the NFEs in both forward and backward passes while improving the negative ELBO on the test set compared to the NODE and GHBNODE.

Furthermore, **we have included another new experiment on the Human Activity dataset [Lustrek et al. (2010)]** in Appendix E.6 of our revision. This dataset consists of time series from five individuals doing various activities: walking, sitting, lying, etc. The task is to classify each time point into one of seven types of activities (walking, sitting, etc.). We use the NODE-RNN, i.e. the ODE-RNN model in [Rubanova et al.(2019)], as the baseline model. We observe that our GNesterovNODE-RNN outperforms the baseline NODE-RNN and GHBNODE-RNN in both test accuracy and NFEs. We summarize our results below and in Figure 18 and Table 8 in Appendix E.6 of our revision.

We have summarized the changes we made in the manuscript in the Summary of Revision below. We would appreciate it if you could let us know if there are additional questions or concerns about our revision and rebuttal. We would be happy to address any additional comments.

Table 1: The FFJORD-GNesterovNODE vs. the baseline FFJORD-NODE and FFJORD-GHBNODE for use in variational inference with a continuous normalizing flow model, i.e. FFJORD [Grathwohl et al., 2019a], on the binarized MNIST dataset. We also include the reported results from [Grathwohl et al., 2019a] (in parentheses) in addition to our reproduced results.

| Method      | Average Forward NFEs  | Average Backward NFEs   | Negative ELBO
| :---        |    :----:   |   :----:   | :----:   |
| FFJORD-NODE    |  97.92   |  100.76  | 88.30 (82.82)  |
| FFJORD-GHBNODE     |  90.33     |  98.77  | 75.87  |
| FFJORD-GNesterovNODE     |    **62.04**   | **51.07**   | **72.16**  |


Table 2: The GNesterovNODE-RNN vs. the baseline NODE-RNN and GHBNODE-RNN on the Human Activity benchmark [Lustrek et al. (2010)].

| Method      | Average Forward NFEs  | Average Backward NFEs   | Test Accuracy (%)
| :---        |    :----:   |   :----:   | :----:   |
| NODE-RNN    |  220.74   |  396.42  | 82.90  |
| GHBNODE-RNN     |  51.85     |  221.26  | 83.80  |
| GNesterovNODE-RNN     |    **46.44**   | **210.84**   | **84.00**  |


**References**

[1] Lustrek, M. et al. Localization data for person activity data set, 2010.

[2] Rubanova, Y. et al. Latent ordinary differential equations for irregularly-sampled time series. NeurIPS, 2019.

[3] Grathwohl, W. et al.. Ffjord: Free-form continuous dynamics for scalable reversible generative models. ICLR 2019.

---

### Meta-Review · Area_Chair_mi6M · 2022-08-26

**Recommendation:** Accept
**Confidence:** Certain

**Metareview:**

The paper proposes an elegant way to use continious analogue for Nesterov accelerated gradient method instead of the Neural Ordinary Differential Equations (NODE) models for mapping the initial conditions to the output. The authors derive the adjoint ODEs, and show that such new "layer" results in lower number of function evaluations while integrating the NesterovODE. Thus, the authors clearly confirmed that the method is practical and efficient in all scenarios  where NODE models are used. This is a good paper.

**Award:**

No

---

### Decision · Program_Chairs · 2022-09-14

Accept